# SCALING BEHAVIOR OF DISCRETE DIFFUSION LANGUAGE MODELS

**Dimitri von Rütte**[*1]**, Janis Fluri, Omead Pooladzandi, Bernhard Schölkopf**[1 2 3]**,**
**Thomas Hofmann**[1]**, Antonio Orvieto**[2 3]
[1]ETH Zürich [2]ELLIS Institute Tübingen [3]Max Planck Institute for Intelligent Systems,
Tübingen [*]Correspondence to: `dvruette@ethz.ch`

## ABSTRACT

Modern LLM pre-training consumes vast amounts of compute and training data, making the scaling behavior, or scaling laws, of different models a key distinguishing factor. Discrete diffusion language models (DLMs) have been proposed as an alternative to autoregressive language models (ALMs). However, their scaling behavior has not yet been fully explored, with prior work suggesting that they require more data and compute to match the performance of ALMs.

We study the scaling behavior of DLMs on different noise types by smoothly interpolating between masked and uniform diffusion while paying close attention to crucial hyperparameters such as batch size and learning rate. Our experiments reveal that the scaling behavior of DLMs strongly depends on the noise type and is considerably different from ALMs. While all noise types converge to similar loss values in compute-bound scaling, we find that uniform diffusion requires more parameters and less data for compute-efficient training compared to masked diffusion, making them a promising candidate in data-bound settings. We scale our uniform diffusion model up to 10B parameters trained for $10^{22}$ FLOPs, confirming the predicted scaling behavior and making it the largest publicly known uniform diffusion model to date. Training code and models are open-source: `https://github.com/dvruette/gidd-easydel`

## 1 INTRODUCTION

Diffusion language models (DLMs) have recently emerged as an alternative to autoregressive language models (ALMs), promising to address some fundamental limitations plaguing ALMs such as the inability to generate multiple tokens in parallel as well as the inability to revise previously generated tokens (Li et al., 2025). While DLMs' performance at small scales lags behind autoregressive models, they have the potential to solve both of these limitations by decomposing the generative process into a sequence of denoising steps where the entire generated sequence of $N$ tokens is gradually refined, starting at pure noise and transforming it to pure signal over the course of $T$ denoising steps. The freedom to choose $T$ independently of $N$ enables the generation of multiple tokens in each step, while also retaining the ability to update every token at every step.

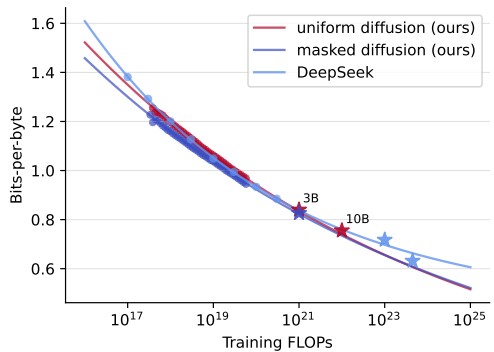

Figure 1: Our proposed scaling laws extrapolate well to 3B and 10B models trained on up to 50× larger compute budgets and suggest that DLMs can be competitive with ALMs at scale, even in compute-bound training settings.[1]

---

[1]Our bpb values are not directly comparable to those of autoregressive models since our value is a mixture of conditional and unconditional likelihoods.

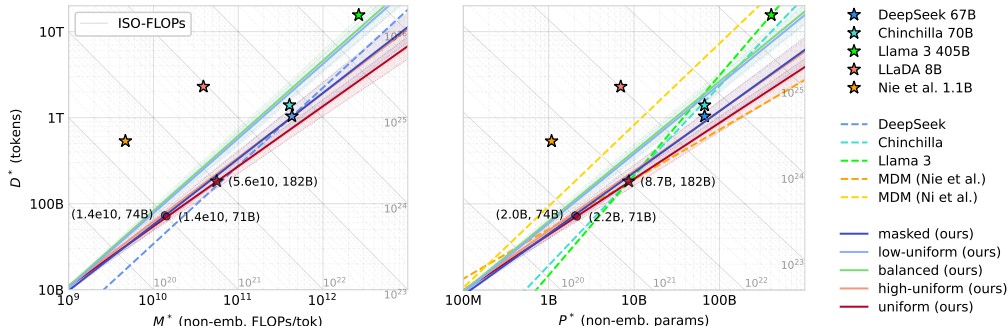

Figure 2: Compute-optimal token-to-parameter ratios as a function of model size can vary significantly for different training objectives: While ALMs generally call for more training tokens relative to parameters, as shown by Chinchilla (Hoffmann et al., 2022), Llama 3 (Grattafiori et al., 2024), and DeepSeek (Bi et al., 2024), discrete diffusion models require comparatively more parameters. For masked diffusion, there is a noticeable disagreement between our results and the literature, with Nie et al. (2025a) predicting more parameter-heavy scaling and Ni et al. (2025) predicting more token-heavy scaling.

Within DLMs, masked diffusion models (MDMs) (Austin et al., 2021; Ou et al., 2025; Sahoo et al., 2024; Shi et al., 2024) have emerged as the predominant DLM archetype next to alternative diffusion processes such as uniform diffusion (Austin et al., 2021; Schiff et al., 2024) or hybrid-noise diffusion (von Rütte et al., 2025). MDMs work by gradually masking tokens and training a model to undo this degradation process by filling in the missing tokens. In contrast, uniform diffusion replaces tokens with random other tokens from the vocabulary until, eventually, every token in the sequence is completely random. Hybrid diffusion models lie on the spectrum between masking and uniform diffusion, utilizing some combination of both noise types. MDMs have gained popularity due to their superior performance at small scales, but face significant challenges despite their dominance. Prior work has suggested that MDMs are less efficient to train, requiring $16\times$ more compute in a compute-optimal setting to match the training loss of ALMs (Nie et al., 2025a). Additionally, like ALMs, MDMs suffer from the inability to revise previously generated tokens. This is due to the fact that every token experiences exactly one state transition (between its masked and unmasked state), hence prohibiting any transitions between two unmasked states. This has prompted the realization that alternative diffusion types were, perhaps, abandoned prematurely.

The likelihood gap between autoregressive, masked diffusion, and uniform diffusion models can be explained, at least in part, through the lens of task difficulty: MDMs are trained to generate the data in any random order, which includes, but is not limited to, generating the data in its natural, autoregressive order and is therefore a strictly more difficult problem (Kim et al., 2025). Similarly, uniform diffusion can be understood as a strictly more difficult version of masked diffusion where the model has to predict which tokens are noisy and which are noise-free in addition to subsequently imputing the noisy tokens[2] (Amin et al., 2025). Put differently, going from autoregression to masking to uniform diffusion imposes progressively less structure on the generative process and therefore provides less inductive bias, suggesting that a more expressive model is required to learn the task effectively. Crucially, the scaling behavior of uniform and hybrid-noise DLMs remains an open question, with existing work being limited to small-scale ablations. Furthermore, prior work on scaling MDMs (Nie et al., 2025a) makes some potentially undesirable design choices, such as assuming that the training loss can approach zero given infinite compute as well as fixing the learning rate and batch size to a constant value.

In this work, we refine the strategy from Nie et al. (2025a) by putting additional care on tuning crucial hyperparameters and comparing both compute- and token-bound scaling laws for different noise types, including masked, uniform and hybrid-noise diffusion. Our contributions are three-fold:

---

[2]To see this, consider a hypothetical uniform diffusion scenario where we are given additional information about which tokens are noisy and which are noise free. In this case, the denoising problem becomes equivalent to masked diffusion as it suffices to fill in the noisy (missing) tokens.

**(1) Diffusion process.** To aid with scaling across different noise types, we propose a new family of hybrid diffusion that allows us to easily and smoothly interpolate between masked and uniform diffusion by defining a transition point from masking to uniform diffusion depending on the signal-to-noise-ratio (SNR). We argue that defining the diffusion process through SNR rather than time is more natural and more principled, having become the standard for continuous-state diffusion (Kingma et al., 2021; Kingma & Gao, 2023; Karras et al., 2024). To derive the ELBO of the proposed diffusion process, we frame it as an instance of generalized interpolating discrete diffusion (GIDD; von Rütte et al., 2025) and reparameterize the GIDD ELBO in terms of SNR. This reparameterization simplifies both theory and implementation, while also closing the gap to continuous-state diffusion theory and showing that interpolating discrete diffusion, like continuous diffusion, is invariant to the noise schedule (Kingma et al., 2021).

**(2) Methodology.** We then systematically analyze the scaling behavior across all noise types (masking, uniform, and hybrid), model sizes, training durations, and batch sizes. To aid with scaling, we utilize CompleteP (Dey et al., 2025) for stable learning rate transfer across model width and depth. Instead of fixing the batch size to a constant value, as is often done in prior work on scaling laws, we find it to be a crucial hyperparameter with an optimal value depending on the training token budget. Thus, it requires careful tuning at each scale, leading us to estimate the scaling laws without learning rate annealing in order to cope with this additional scaling dimension. This is motivated by the recent trend of treating pre-training and annealing as two distinct training stages conducted on potentially different datasets (Project Apertus, 2025; Allal et al., 2025), as well as our own ablations showing that training with and without annealing yields similar optima and a similar loss (up to some constant factor).

**(3) Scaling behavior.** The discovered scaling laws paint a picture that is generally favorable for DLMs: Not only do diffusion models, and uniform diffusion in particular, incentivize more parameter-heavy scaling compared to ALMs, making them more token-efficient at compute-optimality, but they also appear to scale competitively in compute-bound settings (Fig. 1). Furthermore, the compute-bound scaling behavior remains largely the same across noise types with no clear advantage or disadvantage for one over the other at scale. To validate our predictions, we scale to 3B parameters (masked and uniform diffusion) and 10B parameters (uniform diffusion) models trained for $10^{21}$ and $10^{22}$ FLOPs respectively, finding that the observed performance closely follows the predicted trend. This makes DLMs, and uniform diffusion in particular, a promising competitor to the predominant autoregressive paradigm, with the potential to match or outperform ALMs at large scales. We also find that the optimal values for batch size and learning rate are remarkably predictable, with the optimal batch size being a function of dataset size, optimal learning rate being a function of (optimal) batch size, and both being largely independent of model size and noise type.

## 2 METHOD

### 2.1 DISCRETE DIFFUSION MODELS

Discrete diffusion models are a special application of diffusion models (Sohl-Dickstein et al., 2015) to discrete data. Conceptually, diffusion models are trained to reverse a corruption process that gradually transforms clean data into pure noise, also referred to as the prior distribution. This allows them to generate novel data by gradually removing noise from some noisy sample, starting at pure noise and gradually moving towards cleaner and cleaner version. Diffusion models define a *forward* noising process $q_{t|s}(z_t|z_s)$ with $0 \leq s < t \leq t$, which is a Markov chain that gradually adds noise to the latent variable $z_s$, starting at $z_0 := x$ and gradually increasing the noise level until eventually, at $t = 1$, the prior distribution $p_1(z_1)$ consisting of pure noise is reached. The denoising model $p_\theta(z_s|z_t)$ is then trained to remove noise by matching the (conditional) *backward* denoising process $q_{s|t}(z_s|z_t, x)$. Discrete diffusion models (Austin et al., 2021) are a special case of this paradigm where the Markov chain operates on a discrete state spaces. In this case, the transitions and marginals of the Markov chain are categorical distribution. For conciseness, we use a vector-based notation for these categorical distributions, writing $\boldsymbol{q}_t(x)$ and $\boldsymbol{q}_{t|s}(z_s)$ to denote the marginals $q_t(\cdot|x)$ and the Markov transitions $q_{t|s}(\cdot|z_s)$ respectively.

## 2.2 GENERALIZED INTERPOLATING DISCRETE DIFFUSION

We adopt generalized interpolating discrete diffusion (GIDD; von Rütte et al., 2025), a class of discrete diffusion models (Austin et al., 2021) that provides a unified perspective of many existing approaches such as masked diffusion (Ou et al., 2025; Sahoo et al., 2024; Shi et al., 2024) or uniform diffusion (Schiff et al., 2024; Sahoo et al., 2025a). GIDD defines the noising process to be an interpolated categorical distribution between some initial state $z_0 := x \in \mathcal{X}$ (the data) and some arbitrary (smoothly time-varying) mixing distribution $\boldsymbol{\pi}_t$ over latent (noisy) variables $z_{t \in (0,1]} \in \mathcal{Z} \supseteq \mathcal{X}$. Specifically, for some mixing rate $\alpha_t$ which determines the signal strength over time, the Markov chain transitions and marginal distributions are given by

$$\boldsymbol{q}_t(x) = \alpha_t \boldsymbol{x} + \beta_t \boldsymbol{\pi}_t, \quad \boldsymbol{q}_{t|s}(z_s) = \alpha_{t|s} \boldsymbol{z}_s + \beta_{t|s} \boldsymbol{\pi}_{t|s} \tag{1}$$

with $\beta_t = 1 - \alpha_t$, $\alpha_{t|s} = \alpha_t/\alpha_s$, $\beta_{t|s}\boldsymbol{\pi}_{t|s} = \beta_t\boldsymbol{\pi}_t - \alpha_{t|s}\beta_s\boldsymbol{\pi}_s$, and $\boldsymbol{x}$ and $\boldsymbol{z}_s$ denoting the one-hot encoding of $x$ and $z_s$ respectively. Under the condition that $\alpha_t$ and $\boldsymbol{\pi}_t$ are differentiable in time, the diffusion negative ELBO (NELBO) of GIDD is given by

$$-\log p_\theta(x) \leq \mathbb{E}_{t \sim \mathcal{U}(0,1), z \sim \boldsymbol{q}_t(x)} \left[ \boldsymbol{w}_t(x)_z \{ D_{KL}(\boldsymbol{q}_t(x) \| \boldsymbol{q}_t(\boldsymbol{x}_\theta)) + D_{IS}(\boldsymbol{q}_t(x)_z \| \boldsymbol{q}_t(\boldsymbol{x}_\theta)_z) \} \right] + C, \tag{2}$$

with $D_{IS}(p\|q) = p/q - \log p/q - 1$ denoting the (point-wise) Itakura-Saito divergence and $\boldsymbol{w}_t(x)$ is a weighting vector defined as follows (with purely element-wise operations):

$$\boldsymbol{w}_t(x) = \frac{1}{\boldsymbol{q}_t(x)} \left( \beta_t \boldsymbol{\pi}_t' - \frac{\alpha_t'}{\alpha_t} \boldsymbol{\pi}_t \right). \tag{3}$$

We adopt the framework of von Rütte et al. (2025) as it allows us to train discrete diffusion models with different noising properties within a shared framework, reducing precisely to specialized variants in the literature under an appropriate mixing schedule. However, we improve this framework by showing how it can be reformulated in terms of signal-to-noise ratio (SNR), obtaining a simpler, more flexible likelihood bound and closing the gap to continuous-state diffusion theory.

## 2.3 REFRAMING GIDD IN TERMS OF SNR

It is well-known that continuous-state diffusion models are invariant to the noise schedule (Kingma et al., 2021), with many approaches relying on this fact to accelerate training via adaptive noise schedules (Kingma & Gao, 2023; Karras et al., 2024; Dieleman, 2024). This stems from the insight that the notion of time in diffusion models is spurious and serves only as a proxy for the signal-to-noise ratio (SNR), and that SNR is sufficient and, arguably, a more natural way to describe the forward and backward diffusion process. Similar results have been shown for the special case of masked diffusion (Shi et al., 2024; Sahoo et al., 2024). In this section, we show that this invariance continues to hold for general interpolating discrete diffusion models following the proof technique by Kingma et al. (2021).

First, we define the log-SNR $\lambda$ as $\lambda = \log \frac{\alpha}{1-\alpha}$, which connects it to the signal strength $\alpha$ via the sigmoid relation $\alpha = \sigma(\lambda)$ where $\sigma(\lambda) = \frac{1}{1+e^{-\lambda}}$. Then, the GIDD forward process (Eq. 1) as a function of $\lambda$ is given simply by

$$\boldsymbol{q}_\lambda(x) = \sigma(\lambda)\boldsymbol{x} + \sigma(-\lambda)\boldsymbol{\pi}_\lambda \tag{4}$$

which is all we need to rewrite the GIDD ELBO in terms of log-SNR.[3] See App. C for the proof.

**Proposition 1.** *The GIDD ELBO (Eq. 2) can be expressed as an importance sampling procedure over log-SNRs $\lambda \sim p(\lambda)$ and the forward noising process $z \sim \boldsymbol{q}_\lambda(z)$.*

$$-\log p(x) \leq \mathbb{E}_{\lambda,z} \left[ \frac{\boldsymbol{w}_\lambda(x)_z}{p(\lambda)} \{ D_{KL}(\boldsymbol{q}_\lambda(x) \| \boldsymbol{q}_\lambda(\boldsymbol{x}_\theta)) + D_{IS}(\boldsymbol{q}_\lambda(x)_z \| \boldsymbol{q}_\lambda(\boldsymbol{x}_\theta)_z) \} \right] + C, \tag{5}$$

*with the weighting term $\boldsymbol{w}_\lambda(x) = \frac{\sigma(-\lambda)(\boldsymbol{\pi}_\lambda - \boldsymbol{\pi}_\lambda')}{\boldsymbol{q}_\lambda(x)}$.*

While the choice of $p(\lambda)$ is arbitrary, we set $p(\lambda) = \sigma'(\lambda)$ in our experiments, which corresponds to the linear noise schedule $\sigma(\lambda) = \alpha = 1 - t$.

---

[3]The conditional transitions in terms of log-SNR, while not needed for the proof, are given by Eq. 1 with $\alpha_{\lambda_t|\lambda_s} = \sigma(\lambda_t)/\sigma(\lambda_s)$ and $\beta_{\lambda_t|\lambda_s}\boldsymbol{\pi}_{\lambda_t|\lambda_s} = \sigma(-\lambda_t)\boldsymbol{\pi}_{\lambda_t} - \alpha_{\lambda_t|\lambda_s}\sigma(-\lambda_s)\boldsymbol{\pi}_{\lambda_s}$.

## 2.4 A Universal Hybrid Mixing Distribution

Hybrid-noise diffusion (Gu et al., 2022; von Rütte et al., 2025; Haxholli et al., 2025) has been proposed as a way to equip masked diffusion models with the ability to revise tokens throughout the denoising process while having a smaller likelihood gap compared to fully uniform diffusion. For our scaling experiments, we consider a mixing distribution $\boldsymbol{\pi}_\lambda$ that smoothly transitions from masked to uniform diffusion, covering a range of hybrid mixtures in between. Our idea is to interpolate between the pure masking and pure uniform noise based on the log-SNR $\lambda$, thereby controlling how much masking and how much random perturbation happens proportionally at any point of the noising process. We define

$$\boldsymbol{\pi}_\lambda = \sigma(a\lambda + b)\boldsymbol{u} + (1 - \sigma(a\lambda + b))\boldsymbol{m}, \tag{6}$$

with $\sigma$ denoting the sigmoid function, $\boldsymbol{u} = \frac{1}{N-1}(1 - \boldsymbol{e}_m)$ and $\boldsymbol{m} = \boldsymbol{e}_m$ denoting the uniform and masking probability vector respectively, and $a, b$ being hyperparameters that control the transition point and speed between masking and uniform noise. Note that for $a > 0$ this mixing distribution approaches pure masking as $b \to -\infty$ and pure uniform noise as $b \to \infty$, with varying masking-to-uniform mixtures in between. In our experiments, we fix $a = 1$ for simplicity. The reparameterized ELBO enables trivial implementation of this mixing distribution, only requiring computation of the derivative of $\boldsymbol{\pi}'_\lambda$, which is given by $\boldsymbol{\pi}'_\lambda = a\sigma'(a\lambda + b)(\boldsymbol{u} - \boldsymbol{m})$.

## 2.5 Anisotropic Noise and Diffusion Forcing

While sequence diffusion models typically use a global *isotropic* noise level that is shared by all tokens in the sequence, Diffusion Forcing (Chen et al., 2024) proposes to sample noise levels independent for each tokens, resulting in *anisotropic* noise. This is done to stabilize autoregressive rollouts and generally provides enhanced flexibility at inference time as it effectively serves as an augmentation over noise levels. This idea has since been extended to DLMs (Wang et al., 2025) to speed up inference.

## 3 Estimating scaling laws

Scaling laws have become an important ingredient of large-scale neural network training, particularly in the context of training LLMs. Due to the vast costs associated with large-scale training runs, key decisions are based on forecasts obtained through extrapolating the performance of smaller runs to the desired, bigger scale. Prior work on the scaling of MDMs (Nie et al., 2025a) has made some assumptions that we would like to revisit. For example, the learning rate and batch size are fixed to constant values across all experiments, but this may not be optimal for different model sizes and token budgets (Bergsma et al., 2025). Additionally, the reported scaling law is the result of a power law fit without constant offset, thereby implicitly assuming that the ideal training loss is zero and can be reached given infinite compute, which is known not to be the case for ALMs. These limitations prompt us to rederive the scaling laws from scratch, dropping any assumptions on the optimal batch size, learning rate, and irreducible loss. Our scaling laws are estimated directly on the negative ELBO,[4] which constitutes an upper bound on the negative log-likelihood (NLL). While our recipe largely follows the methodology by (Hoffmann et al., 2022), which is well-established and has been widely adopted for estimating scaling laws (Touvron et al., 2023; Bi et al., 2024; Shuai et al., 2024), there are some key differences.

**Maximal update parameterization.** To aid with the scaling process, we adopt CompleteP (Dey et al., 2025), a variant of $\mu$P (Yang et al., 2022) that parameterizes the model such that optimal learning rates transfer both across width and depth. Unlike the original work, we do not employ a base width to keep learning dynamics equivalent to some reference model and instead find the optimal values for weight initialization variance and base learning rate through a hyperparameter sweep on a 25M and 50M parameter model. This results in different optimal values for width-dependent parameters (bulk parameters) such as weight matrices compared to non-width-dependent parameters such as layer-normalization and bias parameters (auxiliary parameters), with bulk parameters requiring a larger initialization variance and learning rate. We find optimal values of $\sigma_{\text{base}} = 0.4$, $\sigma_{\text{aux}} = 0.02$ and $\eta_{\text{base}} = 0.3$, $\eta_{\text{aux}} = 0.02 \cdot \eta_{\text{base}}$ for initialization variances and learning rates respectively (at

---

[4]In a slight abuse of terminology, we sometimes refer to the negative ELBO simply as the "ELBO".

a batch size of $64$). These values transfer remarkably well in our experiments, with only the base learning rate $\eta_{\text{base}}$ requiring adjustments depending on the batch size.

**Learning rate annealing.** While adopting CompleteP enables learning rate transfer across model scales, the same learning rate is not optimal for different batch sizes and training horizons. It is therefore still necessary to sweep the learning rate for each model and batch size in order to find the compute-optimal Pareto frontier. To cope with the computational demands of sweeping the batch size in addition to model and data size, we omit learning rate annealing and analyze the scaling behavior without it. This allows capturing all possible training horizons in a single run per model, batch size, and learning rate. This decision is justified twofold: First, modern large-scale training often treats the annealing phase as distinct from pre-training where the data mixture is often adapted to more closely resemble the test distribution by injecting more high-quality data geared towards the desired downstream tasks (Project Apertus, 2025; Allal et al., 2025). Second, we conduct small-scale ablations to study the effect of omitting annealing, finding that optimal hyperparameters are preserved and that the performance difference is a constant factor (see Sec. 4.4).

**Optimal batch size.** Sweeping the learning rate across batch and model sizes reveals the clear existence of a compute- and token-optimal batch size that scales almost linearly in the number of training tokens (Fig. 3). Similar findings have been reported for ALMs when training below the *critical batch size* (Hu et al., 2024; Shuai et al., 2024; Bergsma et al., 2025). The critical batch size refers to the phenomenon where scaling the batch size past a certain *critical* point yields diminishing returns and becomes compute-inefficient. It is worth noting that we find no dependence of the optimal batch size on the target loss, a claim that has been raised for the critical batch size of ALMs (Zhang et al., 2024) and also more generally (McCandlish et al., 2018). As our experiments show no signs of saturation even at batch sizes at $10^6$ tokens, this suggests that the critical batch size of DLMs lies well above that of ALMs, which has been reported to saturate around $10^6$ tokens (Shuai et al., 2024; Zhang et al., 2024). Another hyperparameter that is known to have optimal values depending on the batch size is Adam's $\beta_2$ parameter. However, also for this parameter we find little benefit in deviating from our default value of $0.99$: Neither do we observe benefits from using larger values for small batch sizes[5] nor from using smaller values at larger batch sizes.[6] We do decrease $\beta_2$ to $0.98$ starting at batch sizes of $256$ as we found this to slightly improve stability without any noticeable performance degradation.

## 4 EXPERIMENTS AND RESULTS

### 4.1 MODEL ARCHITECTURE AND TRAINING

**Architecture.** Our model architecture follows a standard Transformer (Vaswani et al., 2017) with some key modifications. As described in Section 3, we implement CompleteP (Dey et al., 2025) for optimal learning rate transfer across width and depth. We use Squared ReLU for MLP activations, as recommended by So et al. (2021). To ensure stable training, we add RMSNorm (Zhang & Sennrich, 2019) layers without bias before each attention and MLP block following LLaMA (Touvron et al., 2023) as well as to both keys and queries, following QK-norm (Naseer et al., 2021; Dehghani et al., 2023). In the same spirit, we also employ attention logit soft-capping (Gemma Team, 2024). Finally, we add attention sinks in the form of attention biases (Sun et al., 2024) to further stabilize training and prevent outlier features (Sun et al., 2024; He et al., 2024).

**Data.** We use Nemotron-CC (Su et al., 2024) without quality filtering as a representative dataset of internet-scale pre-training. Since it is known that a larger vocabulary facilitates better scaling (Takase et al., 2024; Huang et al., 2025) and to ensure efficient tokenization, we train a BPE tokenizer (Gage, 1994; Sennrich et al., 2015) with a vocabulary size of $2^{17}$ (131,072) tokens on a 256 GB subset of the data. The trained tokenizer is released with the model.

**Diffusion process.** We use the mixing distribution proposed in Section 2.4 with shift $b \in \{-1000, -2, 0, 2, 1000\}$, resulting in pure masking and pure uniform noise for $b = -1000$ and

---

[5]This finding is particular to half-precision training in `bfloat16`, as we did observe slight benefits from increasing $\beta_2$ for full-precision training at low batch sizes.

[6]This improved consistency may be a result of using the LaProp (Ziyin et al., 2020) variant of Adam, or due to the CompleteP (Dey et al., 2025) parameterization, although we do not investigate this further.

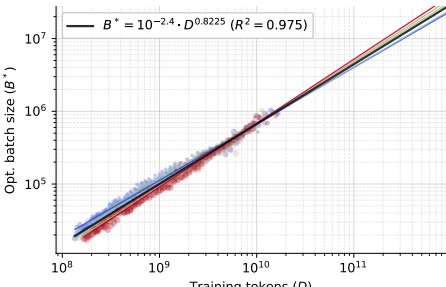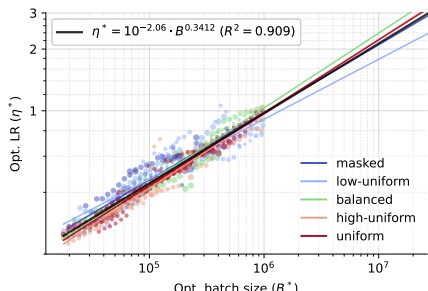

Figure 3: (left) The optimal batch size $B^*$ of discrete diffusion model scales as a power law of training tokens rather than target loss, training FLOPs, or model size. Furthermore, the scaling is close to linear with an exponent of $0.82$. (right) The optimal learning rate $\eta^*$ also follows a power law in batch size (assuming that batch size is optimal) with a scaling exponent of $0.34$.

$b = 1000$ respectively, and hybrid noise with transition points at $t \in \{0.12, 0.5, 0.88\}$, which we refer to as low-uniform, balanced, and high-uniform noise respectively. To balance training stability and ELBO tightness, we restrict the log-SNR to $\lambda \in [-9, 9]$.

We design the diffusion process by aiming to maximize flexibility of the resulting model at inference time, supporting conditional prompt completion, advanced sampling algorithms, as well as flexible length generation. For conditional prompt completion, we select 20% of samples and leave the first $[N \cdot \arccos(r)], r \sim \mathcal{U}(0, 1)$ tokens noise-free. Attention from prompt queries to completion keys is masked in order to enable KV-caching of the prompt during inference. To support both isotropic and anisotropic denoising, we implement diffusion forcing (Chen et al., 2024) by sampling independent per-token noise levels for 50% of samples. Finally, we augment the context with a random fraction $f \sim \mathcal{U}(0, 0.2)$ of empty tokens following Wu et al. (2025) to add some flexibility to the length of generated samples.

**Optimization.** Instead of directly minimizing the ELBO, we use the unweighted ELBO (Eq. 5 with $p(\lambda) := 1$) as a surrogate loss, as this has been found to give better convergence for both hybrid and masked diffusion models (von Rütte et al., 2025; Sahoo et al., 2025b). Following Hafner et al. (2023), we use LaProp (Ziyin et al., 2020) over Adam for its improved stability on a wider range of $\beta_2$ and $\epsilon$ values. The learning rate is warmed up over the first 2000 steps of training and held constant, with most experiments not including a cooldown phase. For the experiments that do have a cooldown phase, we anneal the learning rate to 0 over the last 20% of training following the WSD schedule (Hu et al., 2024; Hägele et al., 2024).

## 4.2 Scaling Laws and Compute-Optimal Frontier

To derive the scaling laws for the proposed class of diffusion models, we train models of five different sizes, ranging from 25M to 570M non-embedding parameters. For each model size, we sweep the learning rate across seven different batch sizes ranging from $2^{14}$ to $2^{20}$ tokens at a sequence length of $N = 2048$ tokens (8 to 512 sequences, spaced by factors of two).

We find that both the optimal batch size and the optimal learning rate follow a very predictable trend (Fig. 3). The optimal batch size appears to depend primarily on the training horizon, with a remarkably strong, almost linear fit in the total number of training tokens. Similarly, the optimal learning rate follows a power law in the optimal batch size, where we assume that the batch size is chosen to be optimal for the given number of training tokens. Recent literature on scaling ALMs has reported similar predictable trends for both batch size and learning rate (Bi et al., 2024; Bergsma et al., 2025). A more detailed analysis of the effect of model size and noise type on optimal hyperparametersis provided in Appendix A.4. Due to the predictability of the optimal learning rate, we sweep between only 2–3 different learning rates around the known optimal values for each batch size. Across all five noise types the resulting grid search spans 510 runs. See App. A.3 for details.

To determine compute-optimal settings for each model size $M$ (in non-emb. FLOPs-per-token) and dataset size (in tokens), we select a set of target FLOPs and scan each observed loss curves for the loss value (in terms of the true ELBO, not the surrogate loss) achieved at the given target FLOPs. For

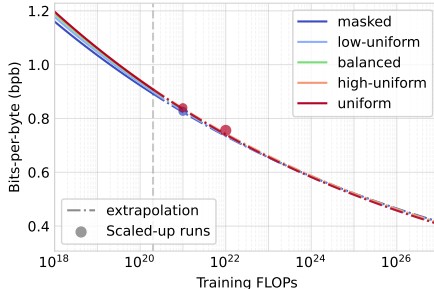 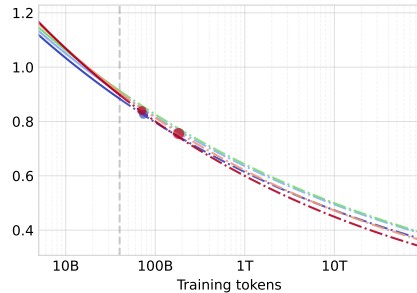

Figure 4: Comparing the scaling laws of different noise types reveals that all noise types approximately converge in the compute-constrained settings (left) while uniform diffusion out-scales other noise types in token-constrained settings (right). Furthermore, the extrapolations remains accurate even up to $50\times$ larger compute budgets than what they were fitted on.

| Model type | $M^* \propto C^{\alpha_M}$ | $D^* \propto C^{\alpha_D}$ | $L^* \propto C^{\alpha_L}$ |
|---|---|---|---|
| *Ours (diffusion)* | | | |
| masked | $0.566_{[-0.022, +0.019]}$ | $0.434_{[-0.019, +0.020]}$ | $-0.0496_{[-0.00041, +0.00032]}$ |
| low-uniform | $0.535_{[-0.012, +0.014]}$ | $0.465_{[-0.014, +0.012]}$ | $-0.0509_{[-0.00034, +0.00035]}$ |
| balanced | $0.534_{[-0.014, +0.015]}$ | $0.466_{[-0.015, +0.014]}$ | $-0.0512_{[-0.00026, +0.00022]}$ |
| high-uniform | $0.573_{[-0.023, +0.027]}$ | $0.427_{[-0.027, +0.023]}$ | $-0.0514_{[-0.00039, +0.00036]}$ |
| uniform | $\mathbf{0.589}_{[-0.019, +0.018]}$ | $\mathbf{0.411}_{[-0.018, +0.0010]}$ | $\mathbf{-0.0522}_{[-0.00029, +0.00026]}$ |
| *Diffusion* | | | |
| MDM (Nie et al., 2025a) | $0.634^\dagger$ | $0.366^\dagger$ | $-0.0615^\dagger$ |
| MDM (Ni et al., 2025) | $0.514$ | $0.486$ | $-$ |
| *Autoregressive* | | | |
| Hoffmann et al. (2022) | $0.49$ | $0.51$ | $-$ |
| Shuai et al. (2024) | $0.464$ | $0.536$ | $-$ |
| Bi et al. (2024) | $0.5243$ | $0.4757$ | $-$ |
| Nie et al. (2025a) | $0.644^\dagger$ | $0.356^\dagger$ | $-0.0633^\dagger$ |

$^\dagger$Scaling coefficients are parsed from the provided plots as the original paper does not report them.

Table 1: We find that *uniform noise* has the best scaling behavior among diffusion models with slightly better loss scaling in compute-bound settings. Furthermore, all diffusion types call for scaling the model size more quickly than ALM scaling laws from the literature.
The scaling laws describe the compute-optimal model size $M^*$ (in non-embedding FLOPs-per-token), training set size $D^*$ (in terms of tokens) and training loss $L^*$ (in terms of ELBO) as a function of training compute $C$ (in non-embedding FLOPs), following the methodology of Bi et al. (2024). The scaling behavior reported by Nie et al. (2025a) and Ni et al. (2025) differs considerably from ours, which could be due to differences in hyperparameters and/or training data. $2\sigma$-confidence intervals based on standard bootstrapping are given as subscripts.

smoothing, we apply a locally linear fit around the closest point to the target loss and determine the FLOP amount at which the target loss is crossed based on the linear fit. To fit the scaling laws, we adopt the approach based on iso-FLOP profiles from Hoffmann et al. (2022) (Approach 2), as we find that the approach based on a parametric loss function (Approach 3) is unreliable and does not fit our data well. To approximate the number of FLOPs-per-token $M$ (as a measure of model expressivity), we follow Bi et al. (2024) and use $M = 6P + 12LDN$ (with non-emb. parameters $P$, layer count $L$, hidden size $D$, sequence length $N$) over to the more crude yet widely used $M = 6P$ approximation.

Remarkably, we find a consistent trend in the scaling behavior of different noise types, with more uniform noise scaling more favorably with increased compute, requiring more parameters and less data to train compute-optimally. This is especially significant as the size of pre-training datasets is beginning to saturate while compute is continuing to become more abundant. Moreover, given that prior work has found comparable scaling behavior between autoregressive and masked diffusion models (Nie et al., 2025a), this suggests that uniform diffusion models have the potential to outscale existing autoregressive training recipes. This finding is consistent with the notion that going from autoregressive modeling to masked diffusion to uniform diffusion imposes progressively less struc-

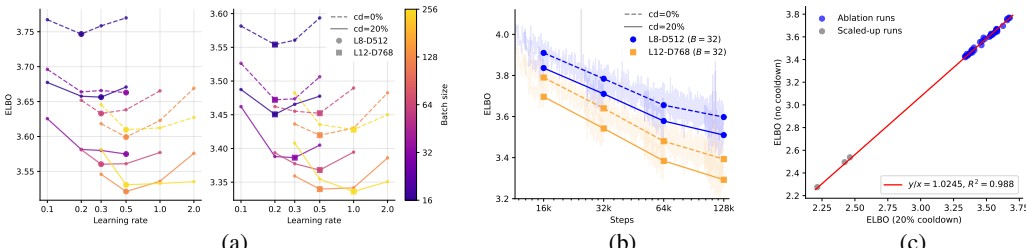

Figure 5: Comparing optimal hyperparameters (a) and different training horizons (b) both without and with 20% learning rate cooldown (abbrv. as 'cd') reveals that learning rate annealing does not affect optimal hyperparameter values and brings a constant improvement across the board. This constant-factor improvement extrapolates remarkably well to even our largest runs (c). Ablations are conducted on 25M and 85M parameter models (denoted as L8-512 and L12-768 respectively).

ture on the generation process and therefore less inductive bias, allowing it to scale more effortlessly with increased compute. Nevertheless, some limitations apply: Scaling coefficients can fluctuate across datasets depending on the data composition (Bi et al., 2024), thus making our numbers not directly comparable with those of Hoffmann et al. (2022) and Shuai et al. (2024).

## 4.3 SCALING UP TO 10B PARAMETERS

Based on our scaling laws and to verify their predictions, we scale up to 3B and 10B parameter models (2.1B and 8.7B non-embedding parameters) trained on $10^{21}$ and $10^{22}$ FLOPs respectively. For the 3B size, we train both a masked and uniform diffusion models, whereas for the 10B size we only train a uniform diffusion model. We find that the observed performance closely matches the predictions (Fig. 4) even for $50\times$ larger runs than what was used to estimate the scaling laws. Importantly, the likelihood gap between masked diffusion and uniform diffusion shrinks from 3.2% at $10^{18}$ FLOPs to only 1.7% at $10^{21}$ FLOPs, supporting the prediction that uniform diffusion requires more capacity to model the data effectively and will eventually catch up. Furthermore, while absolute loss values are difficult to compare between different datasets and tokenizers, we find that our 10B uniform diffusion model matches the scaling trend of DeepSeek (Bi et al., 2024), which uses an autoregressive architecture (Fig. 1). This suggests that DLMs may be competitive with ALMs at scale, a trend that is corroborated by Nie et al. (2025b). Downstream performance of our models, as measured by a range of standard NLP benchmarks, is reported in Appendix A.1.

## 4.4 ABLATING THE EFFECT OF LEARNING RATE ANNEALING

For the sake of reducing the computational burden of scaling law estimation, we use a warmup-stable learning rate schedule without annealing, as outlined in Section 3. While we argue that this is a principled choice due to modern pre-training recipes spending most of the steps in the constant-LR regime and often treating the annealing as a separate phase (Project Apertus, 2025; Kimi Team, 2025; Allal et al., 2025), we empirically investigate the the effect of learning rate annealing and find that it brings a constant improvement of 2.45% ± 0.138%[7] and does not affect optimal hyperparameters. Specifically, we investigate two settings: First, we fix the token budget while varying the batch size, sweeping the learning rate for each batch size both with and without annealing (Fig. 5a). This reveals that both learning rate and batch size have stable optima that are largely unaffected by annealing, which only shifts the final loss by a constant factor. Second, we investigate how the annealed performance evolves over the course of a single training run (Fig. 5b), again finding that the shape of the annealed loss closely matches the unannealed trajectory. Finally, we plot and extrapolate the correlation between the annealed and unannealed loss (Fig. 5c), revealing that the constant improvement continues to hold even for our scaled-up 3B and 10B parameter runs.[8] We therefore conclude that all scaling laws can safely be estimated without learning rate annealing, significantly reducing the number of ablation runs and compute requirements. The constant 2.45% factor can easily be incorporated at any point to account for the effect of adding annealing during the final training.

---

[7]99% confidence interval, obtained through normal approximation.

[8]The unannealed losses for the scaled-up runs are obtained through log-log extrapolation to the final 20% of steps based on the most recent steps before annealing begins.

## 4.5 RELATION BETWEEN BATCH SIZE AND STEP COUNT

While the optimal batch size depends primarily on the total number of training tokens, we addition-ally observe a tight relationship between batch size and step count along iso-loss curves. Concretely, for any target loss $L$, if we iterate through all batch sizes $B$ and observe the step count $S$ at which the target loss is reached (assuming that the model size is fixed and the learning rate well-tuned), then the obtained points can be accurately described by the following equation:

$$\left(\left[\frac{S}{S_{\min}}\right]^{\alpha} - 1\right)\left(\left[\frac{B}{B_{\min}}\right]^{\alpha} - 1\right) = 1, \tag{7}$$

which describes a hyperbola with asymptotes at $S_{\min}$ and $B_{\min}$ and a "stiffness" term $\alpha$ that controls how quickly the curve approaches these asymptotes. This suggests that for a fixed model size there is a minimum step count and minimum batch size required to reach a certain target loss, even when the learning rate is tuned. Within the range of losses we study, this implies that, for a fixed model size and target loss, there is an effective minimum step count and minimum batch size: Using fewer steps than $S_{\min}(L)$ or smaller batches than $B_{\min}(L)$ does not reach the same loss, even when the learning rate is tuned. Minimizing the total number of tokens $D = BS$ subject to Eq. 7 further yields a token-optimal pair $(B^*, S^*)$,

$$B^* = 2^{1/\alpha} B_{\min}, \qquad S^* = 2^{1/\alpha} S_{\min}, \qquad D^* = 4^{1/\alpha} B_{\min} S_{\min}. \tag{8}$$

Note that $\alpha$ is a measure of sensitivity to suboptimal batch sizes, with higher values implying higher sensitivity. In our experiments, $\alpha(L)$ remains relatively constant across target losses (typically in the range 0.1–0.2), while $S_{\min}(L)$ and $B_{\min}(L)$ both increase as we target smaller losses. Over the range of losses we observe, a simple power-law fit provides a convenient summary of this trend (App. A.2), but we do not expect this relation to remain accurate all the way to the irreducible loss. Here it is used purely as a phenomenological description of the regime we actually probe.

At first sight, the mere existence of a positive $B_{\min}(L)$ and of a token-optimal batch size might appear to contradict classical results that advocate for small batches (Keskar et al., 2016; Masters & Luschi, 2018; Smith et al., 2020). However, those analyses typically assume a fixed dataset, arbi-trarily many passes over the data, and focus on regularization effects of gradient noise which helps improve generalization. By contrast, our setting is internet-scale pre-training with an effective num-ber of epochs below one and no observed overfitting. This difference in assumptions resolves the apparent tension with classical small-batch generalization results and is consistent with recent stud-ies in ALM pre-training, which also report a token-dependent optimal batch size and diminishing returns beyond a critical batch size (Hu et al., 2024; Shuai et al., 2024; Bergsma et al., 2025).

## 5 CONCLUSION

We have presented a comprehensive study of scaling laws of discrete diffusion language models, comparing different noise types ranging from masking to uniform noise and paying careful attention to crucial hyperparameters such as learning rate and batch size. The discovered scaling laws paint a favorable picture for both masked and uniform DLMs. We find that all examined DLM variants scale comparatively in compute-bound settings, with uniform diffusion scaling most favorably in token-bound environments. Overall, the scaling of DLMs is competitive with autoregressive models, matching their performance at scale and retaining the potential to overtake them at very large scales thanks to a smaller irreducible loss term. Despite uniform diffusion performing subpar to masked diffusion at small scales, a shrinking likelihood gap along with more a parameter-heavy compute-optimal scaling law supports the hypothesis that uniform diffusion imposes less of an inductive bias on the generative process and needs more capacity to model the data effectively.

Our findings support the case for discrete diffusion language models (DLMs) as a viable alternative to autoregressive language models (ALMs), the prevalent paradigm. DLMs can resolve core lim-itations of ALMs, enabling parallel generation for improved throughput, possessing the ability to revise and self-correct previously generated tokens, providing trivial ways of scaling test-time com-pute, and now also showing signs of improved scaling behavior with increased training compute. All in all, we conclude that DLMs in general, and uniform diffusion in particular, are promising candidates for next-generation LLMs.

## REPRODUCIBILITY STATEMENT

In order to facilitate transparency and reproducibility of our results, we release all of our training code as well as the code used for fitting the obtained scaling laws. Trained model weights are also released along with intermediate checkpoints.

## ETHICS STATEMENT

This paper presents work whose goal is to advance the technical state-of-the-art in an area of Machine Learning. It shares potential societal consequences with much of the work in the general area of language modeling and foundation models.

## ACKNOWLEDGMENTS

This research was supported with Cloud TPUs from Google's TPU Research Cloud (TRC). We thank Erfan Zare Chavoshi for the development of and technical support related to the EasyDeL framework (Zare Chavoshi, 2023), which was used for training our models. Antonio Orvieto and Bernhard Schölkopf acknowledge the financial support of the Hector Foundation. Antonio Orvieto acknowledges the support from the AI2050 Early Career Fellowship by Schmidt Sciences.

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

| Model | Train. FLOPs | ARC-E | ARC-C | WinoG | PIQA | OBQA | BoolQ | GSM8k |
|---|---|---|---|---|---|---|---|---|
| Masking 3B | $10^{21}$ | 49.9 | 29.4 | 51.6 | 64.8 | 30.6 | 60.9 | 1.67 |
| Uniform 3B | $10^{21}$ | 50.6 | 29.4 | 51.1 | 63.5 | 28.8 | 56.4 | 2.05 |
| Uniform 10B | $10^{22}$ | **61.8** | **35.7** | **55.5** | **66.3** | **32.8** | **60.3** | **2.43** |

Table 2: Downstream performance of our scaled models generally correlates with ELBO, with the 3B masked model slightly outperforming the 3B uniform model on average.

| Model | Ancestral ($T = 256$) | Adaptive ($T = 128$) | Adaptive ($T = 256$) |
|---|---|---|---|
| Masking 3B | 1.06 | **1.67** | – |
| Uniform 3B | 1.44 | 1.97 | **2.05** |
| Uniform 10B | 2.12 | 2.27 | **2.43** |

Table 3: Confidence-based, or adaptive, sampling improves accuracy on GSM8k noticeably for all models. Furthermore, uniform diffusion outperforms masked diffusion in any setting and is able to further improve accuracy by investing more denoising steps $T$.

## A  ADDITIONAL RESULTS

### A.1  DOWNSTREAM EVALUATIONS

We report the downstream performance according to a range of standard NLP benchmarks in Table 2. While the overall performance correlates with training ELBO, there appears to be a slight pattern of uniform diffusion performing comparatively better on reasoning-heavy tasks (ARC-E, ARC-C, GSM8k) and masked diffusion performing slightly better on knowledge-heavy tasks (PIQA, OBQA, BoolQ). The comparatively poor performance on GSM8k can be explained by the fact that our pre-training data is purely based on Nemotron-CC (Su et al., 2024) and contains no dedicated math or coding data. This explanation is corroborated by the fact that all three checkpoints have a 0% pass rate on HumanEval (Chen, 2021). The benchmarks were conducted using the `lm-evaluation-harness`[9] and the considered benchmarks are ARC-E, ARC-C (Clark et al., 2018), WinoGrande (Sakaguchi et al., 2021), PIQA (Bisk et al., 2020), OpenBookQA (Mihaylov et al., 2018), BoolQ (Clark et al., 2019), and GSM8k (Cobbe et al., 2021). Except for GSM8k, these benchmarks are multiple-choice questions where the answer is selected based on likelihood. We report the best accuracy between 128/256 denoising steps and fill any unused context with tokens sampled from the prior distribution.

For GSM8k, we use a confidence-based sampling algorithm, similar to what is often done for masked diffusion (Nie et al., 2025b; Kim et al., 2025). To this end, we extend confidence-based sampling to uniform diffusion and propose a generalized confidence heuristic that is applicable to both masked and uniform diffusion models. Specifically, in each step we fully denoise the top $k = 1$ position that maximizes

$$\text{conf}(z_t) = p_{\text{prior}}(z_t) \cdot (\max_{z'} p_\theta(x = z'|z_t) - p_\theta(x = z_t|z_t)). \tag{9}$$

In the case of masked diffusion, this simplifies to the standard MDM confidence heuristic, which is $\text{conf}(z_t) = \delta_{z_t, m} \max_{z'} p_\theta(x = z'|z_t)$. Intuitively, this selects tokens that are still noisy (based on $p_{\text{prior}}(z_t)$) but also where the model is confident in some token that is different from the current token, i.e. where there is a big potential improvement (based on $\max_{z'} p_\theta(x = z'|z_t) - p_\theta(x = z_t|z_t)$). Applied to GSM8k, this gives a noticeable boost compared to classic ancestral sampling as shown in Table 3. We use a completion length of 128 and fill the remaining context with random tokens sampled from the prior. Note that for uniform diffusion, we are able to use more adaptive denoising steps than there are tokens in the completion out-of-the-box. This is impossible for masked diffusion without remasking, as it is forced to fully commit to at least one token in every step.

---

[9]https://github.com/EleutherAI/lm-evaluation-harness

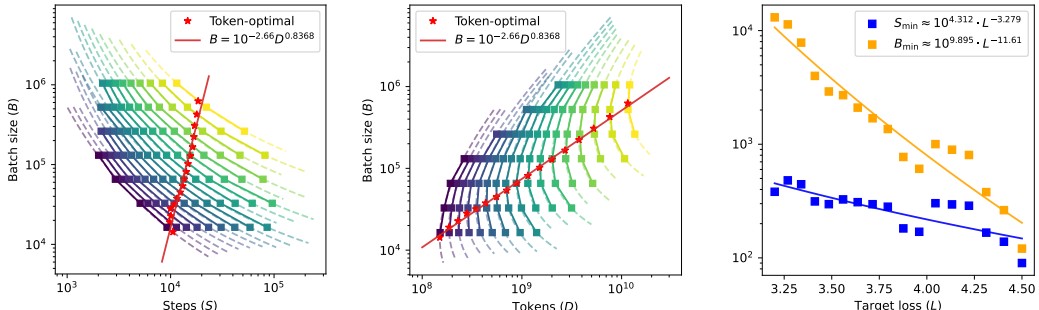

Figure 6: (left, middle) There appears to be a tight relationship between batch sizes and step counts achieving the same loss, with iso-loss curves following a hyperbolic relation. (right) The minimum batch size and step count, as per the asymptotes of the fitted hyperbolas, grow with what appears to be a power law in the target loss, implying that as we get closer to the minimum achievable loss, the minimum required step count, but especially the minimum batch size, grow to large values. Here we display runs of a 85M (L12-D768) model trained on balanced hybrid noise.

| Label | Params. $P$ | Vocab. size $V$ | Layers $L$ | Hidden size $d$ | Attn. heads $H$ |
|---|---|---|---|---|---|
| L8-D512 | 25.2M | 131072 | 8 | 512 | 8 |
| L10-D640 | 49.2M | 131072 | 10 | 640 | 10 |
| L12-D768 | 85.1M | 131072 | 12 | 768 | 12 |
| L16-D1024 | 201.6M | 131072 | 16 | 1024 | 16 |
| L20-D1536 | 566.7M | 131072 | 20 | 1536 | 12 |

Table 4: Overview of the five different model sizes that were used in our experiments. Parameter counts refer to non-embedding parameters.

## A.2 RELATION BETWEEN BATCH SIZE AND STEP COUNT

We given an example of the discovered hyperbolic relation between batch size and step count in Figure 6.

## A.3 SWEEP CONFIGURATION

The exact configurations used for model sizes are given in Table 4 and the hyperparameter sweep settings are given in Table 5. We run smaller batch sizes for $10^5$ optimizer steps, while reducing the number of steps to $5 \times 10^4$ starting at a batch size of 256 sequences for the sake of saving compute.

## A.4 OPTIMAL HYPERPARAMETERS

In Table 6 we report the scaling behavior of optimal hyperparameters (batch size and learning rate) of different noise types and model sizes individually in order to test whether optimal values depend on either of these quantities. We report the slope of a linear fit on a log-log scale along with its $R^2$ value as a measure of goodness of fit. While there are slight fluctuations in the scaling exponents grouped by model size, there is no consistent pattern for both optimal batch size and learning rate, with 99% confidence intervals often overlapping. More data would be needed to form a definitive answer, but based on the available observations we argue that the optimal batch size and learning rate are, more likely than not, unaffected by model size. Note that the independence of learning rate to model size is an expected consequence of using CompleteP (Dey et al., 2025). On the other hand, grouping by noise type reveals a consistent pattern of more uniform noise favoring larger batches. Nevertheless, the difference is rather small and unlikely to have practical implications as batch sizes are commonly rounded to a power of two for computational efficiency. The fits split by noise type are also visualized in Figure 7 (optimal batch size) and Figure 8 (optimal learning rate).

While the optimal batch size does not show any signs of saturation in our experiments, there likely exists a critical batch size above which increasing the size of the batch yields diminishing returns.

| Parameter | Values |
|---|---|
| Noise type $b$ | $\{-1000, -2, 0, 2, 1000\}$ |
| Sequence length $N$ | 2048 |
| LaProp $\beta_1$ | 0.9 |
| LaProp $\beta_2$ | 0.99; 0.98 for $B \geq 256$ |
| LaProp $\epsilon$ | $10^{-8} d^{-1} L^{-1}$ (CompleteP) |
| Init. std. $(\sigma_{\text{bulk}}, \sigma_{\text{aux}})$ | $0.4 \cdot d^{-\frac{1}{2}}, 0.02$ (CompleteP) |
| Resid. multiplier | $4.0 \cdot L^{-1}$ (CompleteP) |
| Out. multiplier | 512 (CompleteP) |
| Weight decay | 0.0 |
| LR warmup steps | 2000 |
| LR cooldown steps | 0 |
| Bulk LR $\eta_{\text{bulk}}$ | $d^{-1} \cdot \eta_{\text{base}}$ (CompleteP) |
| Auxiliary LR $\eta_{\text{aux}}$ | $0.02 \cdot \eta_{\text{base}}$ |
| Param. precision | `bfloat16` |
| Activation precision | Manual mixed precision (BF16, FP32) |
| Batch size $B$ / | 8 / $\{0.2, 0.3\}$ |
| base learning rate $\eta_{\text{base}}$ | 16 / $\{0.2, 0.3, 0.5\}$ |
| | 32 / $\{0.2, 0.3, 0.5\}$ |
| | 64 / $\{0.3, 0.5, 1.0\}$ |
| | 128 / $\{0.5, 1.0\}$ |
| | 256 / $\{0.5, 1.0\}$ |
| | 512 / $\{0.5, 1.0, 2.0\}$ |

Table 5: List of key hyperparameters for our grid search. The parameters that are swept over are noise type, model size (Tab. 4), batch size, and learning rate.

| *Optimal batch size $B^*$* | | | *Optimal learning rate $\eta^*$* | | |
|---|---|---|---|---|---|
| Group | Slope ($\pm 99\%$ CI) | $R^2$ | Group | Slope ($\pm 99\%$ CI) | $R^2$ |
| all | $0.8225_{\pm 0.0104}$ | 0.9754 | all | $0.3412_{\pm 0.0110}$ | 0.9089 |
| masked | $0.7759_{\pm 0.0167}$ | 0.9846 | masked | $0.3295_{\pm 0.0286}$ | 0.8846 |
| low-uniform | $0.7916_{\pm 0.0194}$ | 0.9811 | low-uniform | $0.2924_{\pm 0.0290}$ | 0.8360 |
| balanced | $0.8373_{\pm 0.0214}$ | 0.9806 | balanced | $0.3653_{\pm 0.0207}$ | 0.9437 |
| high-uniform | $0.8607_{\pm 0.0206}$ | 0.9832 | high-uniform | $0.3628_{\pm 0.0190}$ | 0.9489 |
| uniform | $0.8787_{\pm 0.0160}$ | 0.9899 | uniform | $0.3554_{\pm 0.0151}$ | 0.9643 |
| 25.2M | $0.7847_{\pm 0.0202}$ | 0.9838 | 25.2M | $0.3422_{\pm 0.0202}$ | 0.9326 |
| 49.2M | $0.8148_{\pm 0.0220}$ | 0.9806 | 49.2M | $0.4606_{\pm 0.0637}$ | 0.8286 |
| 85.1M | $0.8395_{\pm 0.0180}$ | 0.9854 | 85.1M | $0.3293_{\pm 0.0193}$ | 0.9235 |
| 201.6M | $0.8127_{\pm 0.0175}$ | 0.9840 | 201.6M | $0.3279_{\pm 0.0158}$ | 0.9449 |
| 566.7M | $0.8365_{\pm 0.0163}$ | 0.9861 | 566.7M | $0.3340_{\pm 0.0327}$ | 0.8729 |

Table 6: Slope and $R^2$ values for optimal batch size vs. training tokens and optimal learning rate vs. (optimal) batch size, grouped by noise type and model size. While model size does not have a strong effect on the optimal batch size, there is a consistent pattern of higher proportions of uniform noise requiring larger batch sizes to reach optimal performance. For the optimal learning rate, neither model size nor noise type appear to have a significant effect.

The existence of such a batch size is well-established for autoregressive models (McCandlish et al., 2018; Shuai et al., 2024) and is typically around $10^6$ tokens. Further, these findings may not generalize to multi-epoch training setups, as we operate in the sub-epoch training regime where overfitting is not a concern. Further research will be required to understand how optimal and critical batch sizes behave in multi-epoch training for both DLMs and ALMs.

In Figure 9 we plot the optimal learning rates for each noise type, model size, batch size, and training horizon (in steps). For each batch size, there appears to be a power-law relations between optimal

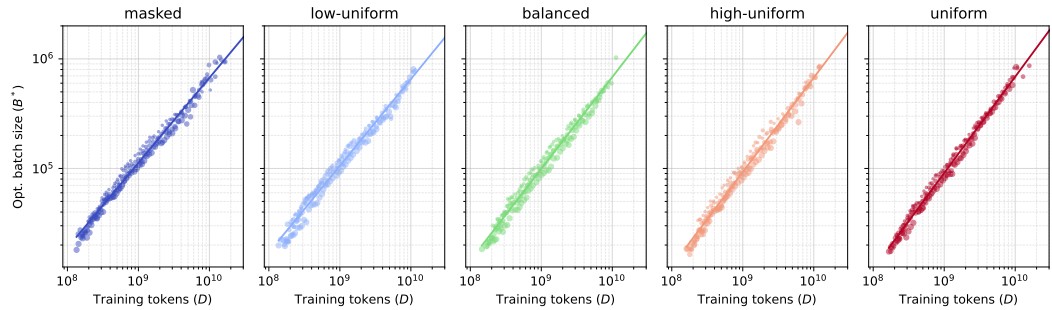

Figure 7: Optimal batch size as a function of training tokens grouped by noise type.

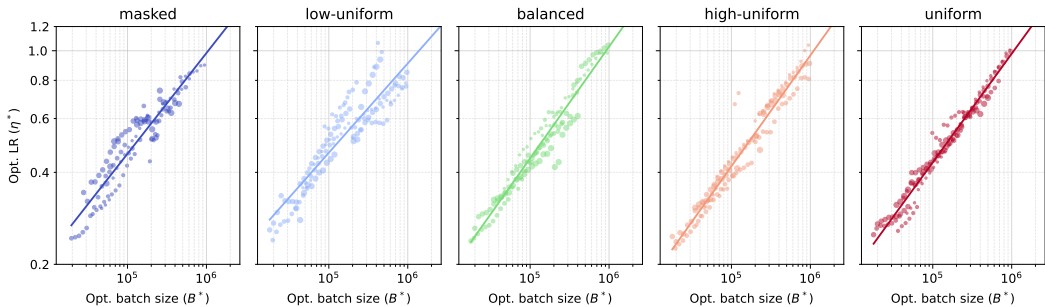

Figure 8: Optimal learning rate as a function of (optimal) batch size grouped by noise type.

learning rate and training steps that is largely independent of model size, which is the expected consequence of using CompleteP (Dey et al., 2025). Furthermore, the optimal learning rate appears to increase with larger batches and decrease with larger training horizons, both of which are expected behaviors.

## A.5 TOKENIZER

Based on a 220 GB sample of the data, our tokenizer uses 4.2278 B/tok (bytes per token), or 0.23653 tok/B, which results in a conversion rate from NLL (in nats) to bpb (bits per byte) as follows:

$$\text{bpb} = 0.34124 \cdot \text{NLL} \tag{10}$$

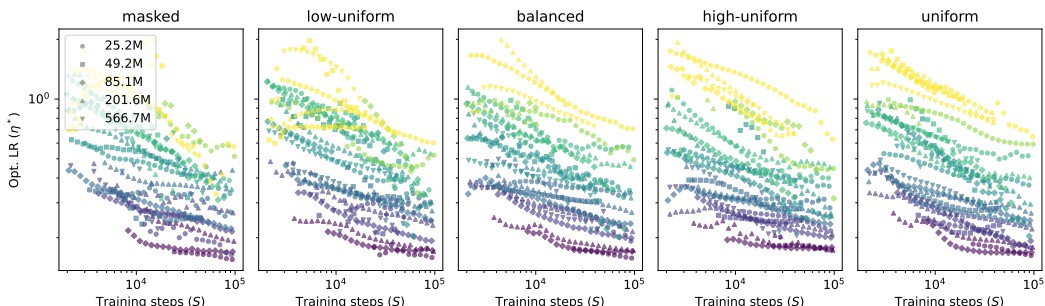

Figure 9: Optimal learning rate $\eta^*$ as a function of training steps split by noise type and colored by batch size (between 8 and 512 sequences, brighter is larger) reveals that $\eta^*$ appears to follow a power law in training steps for each batch size. Model sizes are indicated by different markers.

## B    ON THE DIFFICULTY OF UNIFORM DIFFUSION

In Section 1, we argue that uniform diffusion is a strictly more difficult version of masked diffusion since assuming access to additional information about which tokens are noisy and noise-free reduces uniform diffusion to exactly masked diffusion. Therefore, removing access to this information can never make the task easier. A similar, more formal argument is raised by Amin et al. (2025), saying that uniform diffusion has to learn the time of transition in addition to the transitions themselves, which is not the case for masked diffusion. The authors claim that this provides an inherent, theoretical advantage of masked diffusion over uniform diffusion, which at first glance may appear to contradict our results. However, we argue that both papers provide compatible findings and explanations: Since uniform diffusion is a strictly more difficult task, it requires more modeling capacity, resulting in a likelihood gap at small scales. Borrowing terminology from Amin et al. (2025), small models struggle to learn both the transition schedule and the transitions themselves, but as we increase the size of the models, this becomes less of an issue since there is enough capacity to learn both. This can help further explain why uniform diffusion calls for scaling model size more quickly than data (compared to masked diffusion) and why the likelihood gap between masked and uniform diffusion shrinks with scale.

## C    PROOF OF PROPOSITION 1

To begin, we define the SNR and log-SNR in terms of the mixing rate (or noise schedule) $\alpha_t$ as this is the quantity that determines the proportion of the data distribution (or signal) that is preserved at any given time $t$. Let

$$\text{SNR} = \frac{\alpha}{1-\alpha} \quad \text{and} \quad \lambda = \log \text{SNR} = \log \frac{\alpha}{1-\alpha}. \tag{11}$$

Notably, this results in $\alpha$ being a sigmoid function of $\lambda$, with

$$\alpha = \sigma(\lambda) = \frac{1}{1+e^{-\lambda}}. \tag{12}$$

We will then perform a change-of-variable on the GIDD ELBO, changing the differential from $dt$ to $d\lambda$. Noting the relation between $\alpha$ and $\lambda$, we can rewrite the time-derivative of $\alpha$ as

$$\alpha'_t = \frac{d\alpha}{dt} = \frac{d\alpha}{d\lambda}\frac{d\lambda}{dt} = \frac{d}{d\lambda}\sigma(\lambda) \cdot \frac{d\lambda}{dt} = \sigma(\lambda)\sigma(-\lambda)\frac{d\lambda}{dt} \tag{13}$$

For $\boldsymbol{w}_t(x)$, we then get

$$\boldsymbol{w}_t(x) = \frac{1}{\boldsymbol{q}_t(x)}\left(\beta_t\boldsymbol{\pi}'_t - \frac{\alpha'_t}{\alpha_t}\boldsymbol{\pi}_t\right) = \frac{1}{\boldsymbol{q}_t(x)}\left((1-\sigma(\lambda))\frac{d\boldsymbol{\pi}_\lambda}{d\lambda}\frac{d\lambda}{dt} - \frac{\sigma(\lambda)\sigma(-\lambda)}{\sigma(\lambda)}\frac{d\lambda}{dt}\boldsymbol{\pi}_\lambda\right)$$

$$= \frac{1}{\boldsymbol{q}_t(x)}\sigma(-\lambda)\frac{d\lambda}{dt}\left(\boldsymbol{\pi}'_\lambda - \boldsymbol{\pi}_\lambda\right). \tag{14}$$

Plugging this into Eq. 2, and abbreviating $E_z(\boldsymbol{p},\boldsymbol{q}) := D_{KL}(\boldsymbol{p}\|\boldsymbol{q}) + D_{IS}(\boldsymbol{p}_z\|\boldsymbol{q}_z)$ yields

$$-\log p(x) \leq \mathbb{E}_{t,z}\left[\boldsymbol{w}_t(x)_z E_z(\boldsymbol{q}_t(x),\boldsymbol{q}_t(\boldsymbol{x}_\theta))\right] + C \tag{15}$$

$$= \int_0^1 dt \frac{d\lambda}{dt}\sum_z \sigma(-\lambda)(\boldsymbol{\pi}'_\lambda - \boldsymbol{\pi}_\lambda)_z E_z(\boldsymbol{q}_t(x),\boldsymbol{q}_t(\boldsymbol{x}_\theta)) + C \tag{16}$$

$$= \int_{\lambda_{\min}}^{\lambda_{\max}} d\lambda \sum_z \sigma(-\lambda)(\boldsymbol{\pi}_\lambda - \boldsymbol{\pi}'_\lambda)_z E_z(\boldsymbol{q}_\lambda(x),\boldsymbol{q}_\lambda(\boldsymbol{x}_\theta)) + C. \tag{17}$$

This reveals that the ELBO is invariant not only to the SNR distribution induced by $p(\lambda) = -dt/d\lambda$ but also to the forward process marginals $\boldsymbol{q}_\lambda(x)$, and that their purpose is to approximate this integral through importance sampling. Accordingly, we can convert this back to an expectation like

$$-\log p(x) \leq \mathbb{E}_{\lambda,z}\left[\frac{\boldsymbol{w}_\lambda(x)_z}{p(\lambda)}\{D_{KL}(\boldsymbol{q}_\lambda(x)\|\boldsymbol{q}_\lambda(\boldsymbol{x}_\theta)) + D_{IS}(\boldsymbol{q}_\lambda(x)_z\|q_\lambda(\boldsymbol{x}_\theta)_z)\}\right] + C, \tag{18}$$

with $\lambda \sim p(\lambda)$, $z \sim \boldsymbol{q}_\lambda(x)$, and $\boldsymbol{w}_\lambda(x)_z = \frac{\sigma(-\lambda)(\boldsymbol{\pi}_\lambda - \boldsymbol{\pi}'_\lambda)_z}{\boldsymbol{q}_\lambda(x)_z}$ denoting the updated weighting term.

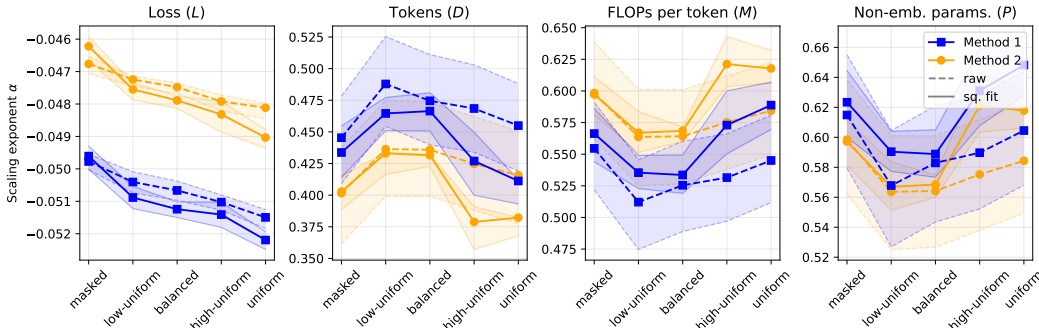

Figure 10: The fitted scaling coefficients differ systematically between FLOP estimation techniques: Method 1 uses the FLOP estimation technique proposed by Bi et al. (2024) whereas method 2 uses the classic approach by Hoffmann et al. (2022). Furthermore, fitting on interpolated data (squared fit) produces tighter confidence bounds and better scaling exponents. Shaded regions denote 95% confidence intervals obtained via standard bootstrapping on the aggregated data points.

## D  SCALING COEFFICIENTS

In this section we provide a complete report of our scaling law estimation methodology along with all fitted coefficients for the different variants. First, in addition to the more accurate FLOP/tok approximation proposed by Bi et al. (2024), $M = 6P + 12LDN$, which we refer to as "Method 1", we also report results for the simpler $M = 6P$ approximation (Kaplan et al., 2020; Hoffmann et al., 2022), which is widely used in the literature (Grattafiori et al., 2024; Nie et al., 2025a; Shuai et al., 2024) and referred to as "Method 2". We also report results both with and without smoothing applied to the iso-FLOP observations (referred to as "raw" and "sq. fit" respectively). It is standard to fit a parabola to observed iso-FLOP losses and taking the minimum thereof (Grattafiori et al., 2024; Bi et al., 2024) as a way of smoothing the observations, and while we find that both approaches qualitatively agree, the low resolution on the model sizes used results in a more brittle fit with broader confidence intervals for the "raw" data. Further, we also ablate whether or not to include an irreducible term in the power law, i.e. whether to use $f(C) = AC^\alpha$ or $f(C) = AC^\alpha + E$ as a hypothesis, and find that, especially for the smoothed data, the irreducible term is typically very small or zero. We therefore conclude that the irreducible term is too small to accurately model, and that omitting it from the hypothesis results in a more robust and comparable fit.

All estimated scaling coefficients, for both FLOP/tok approximations and both with/without data smoothing, are reported in Table 7. Figure 10 reports the scaling exponents with their 95% confidence. Table 8 reports the goodness of fit for all scaling laws in terms of their $R^2$ values. Figures 11 and 12 visualize all data points and the fitted scaling trends along with 95% confidence intervals for the smoothed and raw data respectively. Finally, Table 9 reports all scaling coefficients fitted to a power-law hypothesis which includes an irreducible term.

Table 7: Compute-constrained scaling coefficients for all noise types, metrics, and methodologies, obtained by fitting the power law $AC^\alpha$ (where $C = MD$) to the observed data. Method 1 uses the FLOP/tok estimation from Bi et al. (2024) while method 2 uses the classic $M = 6P$ approximation (Hoffmann et al., 2022). 'raw' interpolation refers to taking the optimal observed value for a given iso-FLOP target (i.e. no interpolation), whereas the 'sq. fit' data is obtained by fitting a parabola to the observed values and taking the optimum thereof. Smallest scaling coefficients are bolded.

| Noise type | Interp. | Method 1 (DeepSeek) | | Method 2 (Chinchilla) | |
|---|---|---|---|---|---|
| Loss ($L$) | | $A$ | $\alpha$ | $A$ | $\alpha$ |
| masked | raw | 27.41772 | -0.04977 | 23.64593 | -0.04676 |
| low-uniform | raw | 28.54100 | -0.05040 | 24.44062 | -0.04724 |
| balanced | raw | 29.11304 | -0.05066 | 24.90152 | -0.04748 |
| high-uniform | raw | 29.69861 | -0.05103 | 25.48828 | -0.04792 |
| uniform | raw | 30.36696 | **-0.05150** | 25.75001 | **-0.04812** |
| masked | sq. fit | 27.21883 | -0.04961 | 23.10227 | -0.04622 |
| low-uniform | sq. fit | 29.12047 | -0.05088 | 24.75090 | -0.04755 |
| balanced | sq. fit | 29.84096 | -0.05124 | 25.33846 | -0.04789 |
| high-uniform | sq. fit | 30.19375 | -0.05141 | 25.92804 | -0.04832 |
| uniform | sq. fit | 31.26171 | **-0.05219** | 26.75888 | **-0.04903** |
| Tokens ($D$) | | $A$ | $\alpha$ | $A$ | $\alpha$ |
| masked | raw | 34.01842 | **0.44542** | 296.71789 | **0.40159** |
| low-uniform | raw | 5.44294 | 0.48788 | 66.84786 | 0.43638 |
| balanced | raw | 9.97715 | 0.47457 | 68.39411 | 0.43586 |
| high-uniform | raw | 12.93277 | 0.46849 | 112.75699 | 0.42477 |
| uniform | raw | 22.48595 | 0.45497 | 171.67127 | 0.41571 |
| masked | sq. fit | 56.78711 | 0.43371 | 294.51814 | 0.40285 |
| low-uniform | sq. fit | 15.31607 | 0.46459 | 80.69196 | 0.43305 |
| balanced | sq. fit | 14.45680 | 0.46642 | 87.87601 | 0.43152 |
| high-uniform | sq. fit | 81.86373 | 0.42701 | 872.33185 | **0.37881** |
| uniform | sq. fit | 161.92545 | **0.41121** | 744.89348 | 0.38224 |
| FLOPs per token ($M$) | | $A$ | $\alpha$ | $A$ | $\alpha$ |
| masked | raw | 0.02940 | 0.55458 | 0.00337 | 0.59842 |
| low-uniform | raw | 0.18373 | **0.51212** | 0.01496 | **0.56362** |
| balanced | raw | 0.10022 | 0.52543 | 0.01461 | 0.56416 |
| high-uniform | raw | 0.07731 | 0.53151 | 0.00887 | 0.57523 |
| uniform | raw | 0.04447 | 0.54503 | 0.00582 | 0.58430 |
| masked | sq. fit | 0.01761 | 0.56629 | 0.00340 | 0.59714 |
| low-uniform | sq. fit | 0.06529 | 0.53541 | 0.01239 | **0.56695** |
| balanced | sq. fit | 0.06918 | **0.53358** | 0.01138 | 0.56848 |
| high-uniform | sq. fit | 0.01222 | 0.57298 | 0.00115 | 0.62119 |
| uniform | sq. fit | 0.00618 | 0.58879 | 0.00134 | 0.61776 |
| Non-emb. params. ($P$) | | $A$ | $\alpha$ | $A$ | $\alpha$ |
| masked | raw | 0.00025 | 0.61488 | 0.00056 | 0.59840 |
| low-uniform | raw | 0.00189 | **0.56784** | 0.00249 | **0.56362** |
| balanced | raw | 0.00095 | 0.58294 | 0.00244 | 0.56413 |
| high-uniform | raw | 0.00071 | 0.58972 | 0.00148 | 0.57523 |
| uniform | raw | 0.00039 | 0.60451 | 0.00097 | 0.58432 |
| masked | sq. fit | 0.00017 | 0.62339 | 0.00057 | 0.59714 |
| low-uniform | sq. fit | 0.00069 | 0.59040 | 0.00207 | **0.56695** |
| balanced | sq. fit | 0.00072 | **0.58881** | 0.00190 | 0.56848 |
| high-uniform | sq. fit | 0.00011 | 0.63110 | 0.00019 | 0.62119 |
| uniform | sq. fit | 0.00005 | 0.64820 | 0.00022 | 0.61776 |

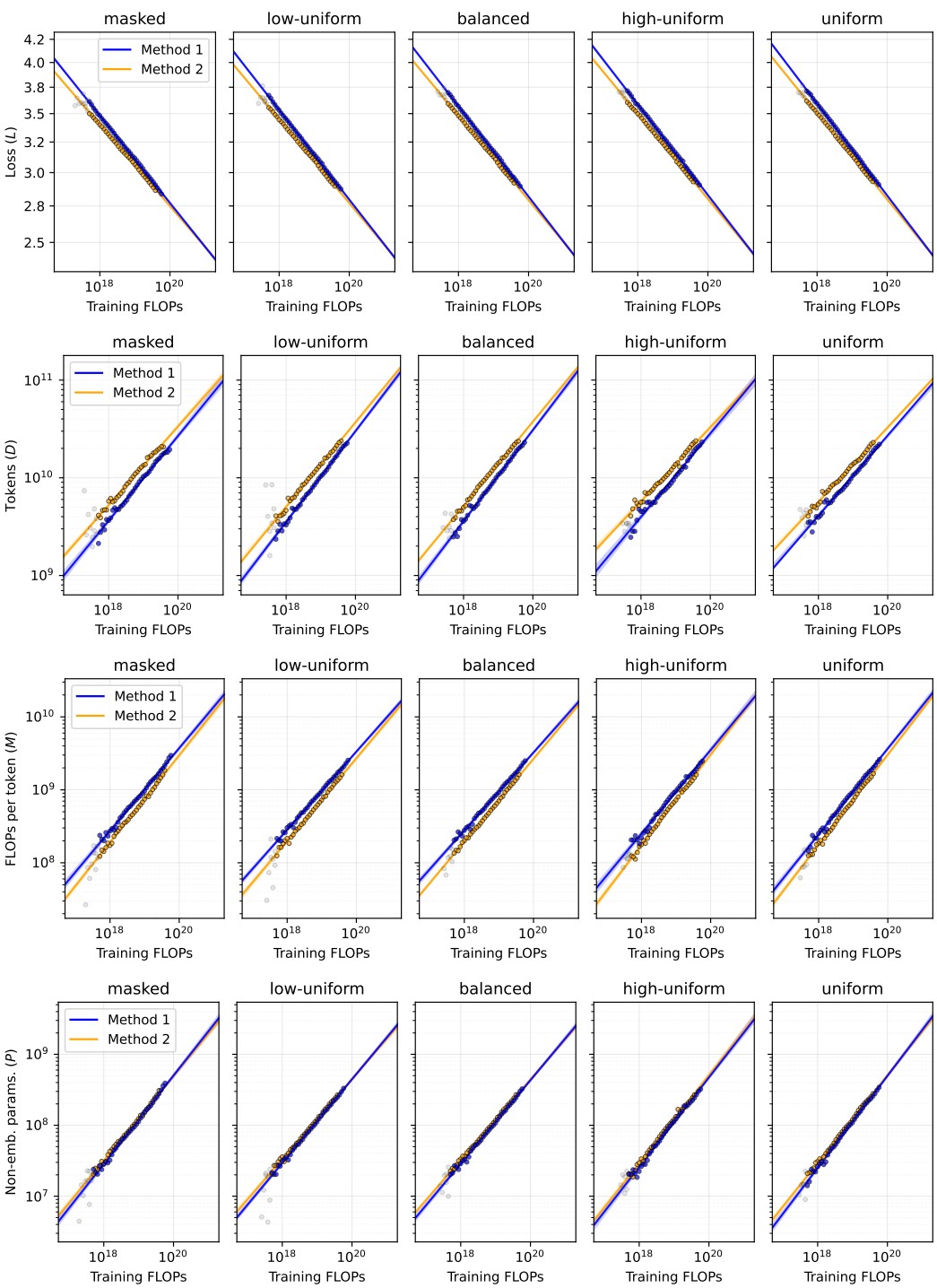

Figure 11: Compute-optimal scaling laws fitted to interpolated values (squared fit). The FLOP estimation methodologies by Bi et al. (2024) (Method 1) and Hoffmann et al. (2022) (Method 2) differ significantly since the FLOP-approximation used by Hoffmann et al. (2022) ($M = 6P$) systematically underestimates the total number of FLOPs executed during training.

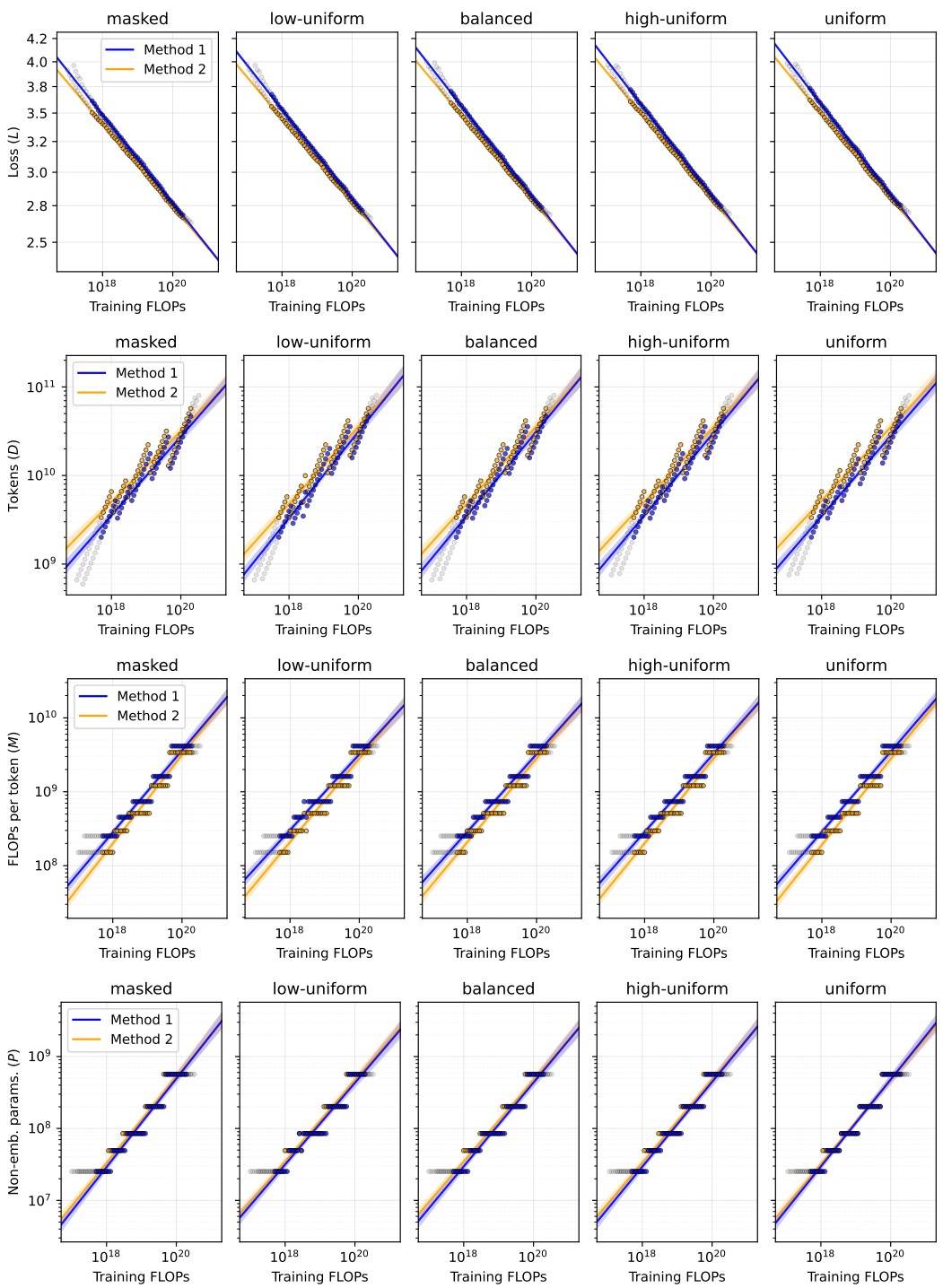

Figure 12: Compute-optimal scaling laws fitted on non-interpolated observations. Compared to interpolated values, the optimal observed values can deviate significantly from the compute-optimal trend due to the small number of unique model sizes used in the sweep. Method 1 uses the FLOP estimation technique from Bi et al. (2024), whereas Method 2 follows Hoffmann et al. (2022).

Table 8: Goodness of fit (as per $R^2$) for all noise types, metrics, and methodologies. Interpolated values ('sq. fit') generally yield a better fit due to the smoothing effect of interpolation, with the exception for the loss, where 'raw' values are already rather smooth.

| Noise type | Method 1 (DeepSeek) | | Method 2 (Chinchilla) | |
|---|---|---|---|---|
| Tokens ($D$) | $R^2$ (raw) | $R^2$ (sq. fit) | $R^2$ (raw) | $R^2$ (sq. fit) |
| masked | 0.9252 | 0.9891 | 0.8898 | **0.9919** |
| low-uniform | 0.9317 | **0.9955** | 0.9086 | 0.9928 |
| balanced | 0.9329 | 0.9938 | 0.9110 | **0.9977** |
| high-uniform | 0.9370 | 0.9810 | 0.9083 | **0.9859** |
| uniform | 0.9344 | 0.9884 | 0.9061 | **0.9926** |
| FLOPs per token ($M$) | $R^2$ (raw) | $R^2$ (sq. fit) | $R^2$ (raw) | $R^2$ (sq. fit) |
| masked | 0.9505 | 0.9936 | 0.9472 | **0.9963** |
| low-uniform | 0.9376 | **0.9966** | 0.9431 | 0.9958 |
| balanced | 0.9446 | 0.9952 | 0.9449 | **0.9987** |
| high-uniform | 0.9504 | 0.9894 | 0.9478 | **0.9947** |
| uniform | 0.9534 | 0.9943 | 0.9502 | **0.9972** |
| Non-emb. params. ($P$) | $R^2$ (raw) | $R^2$ (sq. fit) | $R^2$ (raw) | $R^2$ (sq. fit) |
| masked | 0.9516 | 0.9940 | 0.9472 | **0.9963** |
| low-uniform | 0.9388 | **0.9971** | 0.9431 | 0.9958 |
| balanced | 0.9458 | 0.9958 | 0.9449 | **0.9987** |
| high-uniform | 0.9515 | 0.9909 | 0.9478 | **0.9947** |
| uniform | 0.9542 | 0.9951 | 0.9502 | **0.9972** |
| Loss ($L$) | $R^2$ (raw) | $R^2$ (sq. fit) | $R^2$ (raw) | $R^2$ (sq. fit) |
| masked | **0.9997** | 0.9996 | 0.9997 | 0.9995 |
| low-uniform | 0.9997 | 0.9997 | **0.9998** | 0.9996 |
| balanced | 0.9996 | 0.9997 | **0.9998** | 0.9997 |
| high-uniform | **0.9998** | 0.9995 | 0.9997 | 0.9993 |
| uniform | 0.9996 | **0.9998** | 0.9995 | 0.9997 |

Table 9: Scaling coefficients with intercept, obtained by fitting the power law $AC^\alpha + E$ to the data. The fits almost always have $E \approx 0$, except for the uninterpolated loss values ('raw'), leading us to conclude that setting $E = 0$ is a valid assumption for our setting.

| Noise type | Interp. | Method 1 (DeepSeek) | | | Method 2 (Chinchilla) | | |
|---|---|---|---|---|---|---|---|
| Loss ($L$) | | $A$ | $\alpha$ | $E$ | $A$ | $\alpha$ | $E$ |
| masked | raw | 27.4178 | -0.0498 | 0.0000 | 23.6408 | -0.0468 | 0.0000 |
| low-uniform | raw | 38.6962 | -0.0622 | 0.5941 | 27.0351 | -0.0515 | 0.2578 |
| balanced | raw | 37.5775 | -0.0606 | 0.5192 | 27.3858 | -0.0515 | 0.2446 |
| high-uniform | raw | 36.2488 | -0.0589 | 0.4227 | 28.4489 | -0.0525 | 0.2750 |
| uniform | raw | 41.1977 | **-0.0631** | 0.5872 | 32.4106 | **-0.0574** | 0.5078 |
| masked | sq. fit | 27.2204 | -0.0496 | 0.0000 | 23.1023 | -0.0462 | 0.0000 |
| low-uniform | sq. fit | 29.4611 | -0.0514 | 0.0311 | 24.7510 | -0.0475 | 0.0000 |
| balanced | sq. fit | 29.8418 | -0.0512 | 0.0000 | 25.3384 | -0.0479 | 0.0000 |
| high-uniform | sq. fit | 30.1938 | -0.0514 | 0.0000 | 25.9281 | -0.0483 | 0.0000 |
| uniform | sq. fit | 31.2618 | **-0.0522** | 0.0000 | 26.7589 | **-0.0490** | 0.0000 |
| Tokens ($D$) | | $A$ | $\alpha$ | $E$ | $A$ | $\alpha$ | $E$ |
| masked | raw | 34.0184 | **0.4454** | 0.0000 | 296.7179 | **0.4016** | 0.0000 |
| low-uniform | raw | 5.4429 | 0.4879 | 0.0000 | 66.8479 | 0.4364 | 0.0000 |
| balanced | raw | 9.9772 | 0.4746 | 0.0000 | 68.3941 | 0.4359 | 0.0000 |
| high-uniform | raw | 12.9328 | 0.4685 | 0.0000 | 112.7570 | 0.4248 | 0.0000 |
| uniform | raw | 22.4859 | 0.4550 | 0.0000 | 171.6713 | 0.4157 | 0.0000 |
| masked | sq. fit | 56.7871 | 0.4337 | 0.0000 | 294.5181 | 0.4029 | 0.0000 |
| low-uniform | sq. fit | 15.3161 | 0.4646 | 0.0000 | 80.6920 | 0.4331 | 0.0000 |
| balanced | sq. fit | 14.4568 | 0.4664 | 0.0000 | 87.8760 | 0.4315 | 0.0000 |
| high-uniform | sq. fit | 81.8637 | 0.4270 | 0.0000 | 872.3318 | **0.3788** | 0.0000 |
| uniform | sq. fit | 161.9254 | **0.4112** | 0.0000 | 744.8935 | 0.3822 | 0.0000 |
| FLOPs per token ($M$) | | $A$ | $\alpha$ | $E$ | $A$ | $\alpha$ | $E$ |
| masked | raw | 0.0294 | 0.5546 | 0.0000 | 0.0034 | 0.5984 | 0.0071 |
| low-uniform | raw | 0.1838 | **0.5121** | 0.0095 | 0.0150 | **0.5636** | 0.0063 |
| balanced | raw | 0.1002 | 0.5254 | 0.0006 | 0.0146 | 0.5642 | 0.0010 |
| high-uniform | raw | 0.0773 | 0.5315 | 0.0072 | 0.0089 | 0.5753 | 0.0000 |
| uniform | raw | 0.0445 | 0.5450 | 0.0000 | 0.0058 | 0.5843 | 0.0000 |
| masked | sq. fit | 0.0176 | 0.5663 | 0.0000 | 0.0034 | 0.5971 | 0.0021 |
| low-uniform | sq. fit | 0.0653 | 0.5354 | 0.0000 | 0.0124 | **0.5669** | 0.0000 |
| balanced | sq. fit | 0.0692 | **0.5336** | 0.0072 | 0.0114 | 0.5685 | 0.0000 |
| high-uniform | sq. fit | 0.0122 | 0.5730 | 0.0000 | 0.0011 | 0.6212 | 0.0003 |
| uniform | sq. fit | 0.0062 | 0.5888 | 0.0000 | 0.0013 | 0.6178 | 0.0034 |
| Non-emb. params. ($P$) | | $A$ | $\alpha$ | $E$ | $A$ | $\alpha$ | $E$ |
| masked | raw | 0.0002 | 0.6149 | 0.2830 | 0.0006 | 0.5984 | 0.0027 |
| low-uniform | raw | 0.0019 | **0.5678** | 0.0046 | 0.0025 | **0.5636** | 0.0016 |
| balanced | raw | 0.0010 | 0.5829 | 0.0020 | 0.0024 | 0.5642 | 0.0010 |
| high-uniform | raw | 0.0007 | 0.5897 | 0.0056 | 0.0015 | 0.5752 | 0.0584 |
| uniform | raw | 0.0004 | 0.6045 | 0.0300 | 0.0010 | 0.5843 | 0.0061 |
| masked | sq. fit | 0.0002 | 0.6234 | 0.0008 | 0.0006 | 0.5971 | 0.0073 |
| low-uniform | sq. fit | 0.0007 | 0.5904 | 0.0074 | 0.0021 | **0.5669** | 0.0071 |
| balanced | sq. fit | 0.0007 | **0.5888** | 0.0031 | 0.0019 | 0.5685 | 0.0068 |
| high-uniform | sq. fit | 0.0001 | 0.6311 | 0.0009 | 0.0002 | 0.6212 | 0.0021 |
| uniform | sq. fit | 0.0001 | 0.6482 | 0.0008 | 0.0002 | 0.6178 | 0.0083 |

