# OpenReview forum: "Scaling Behavior of Discrete Diffusion Language Models"
_ICLR.cc/2026/Conference — ICLR 2026 Poster_

### Official Review · Reviewer_foon · 2025-10-30

**Soundness:** 2
**Presentation:** 2
**Contribution:** 2
**Rating:** 4
**Confidence:** 3

**Summary:**

This paper studies the scaling laws of discrete diffusion models. It systematically analyzes the scaling behavior across different noise types (masking, uniform, and hybrid), model sizes, training durations, learning rates (with and without annealing) and batch sizes. In addition, it reformulates Generalized interpolating discrete diffusion (GIDD) as a function of signal-to-noise ratio instead of time.

**Strengths:**

The topic is an important one since indeed as compute becomes more abundant, it is important to know whether diffusion models can scale better than AR ones.

The reformulation of Generalized interpolating discrete diffusion (GIDD) as a function of signal-to-noise ratio instead of time, while not surprising, is valuable.

The finding that annealing of the learning rate moves the ELBO by a constant value is interesting.

**Weaknesses:**

**Main Weaknesses**

*W1* The greatest weakness is that one of the main claims of the paper is that uniform diffusion scales better with compute, but such a statement is not supported empirically. The curves in Figure 4 are extrapolated and will not necessarily behave in that way. In order to make such claims, the author should spend at least 15 times more flops for at least the masked and uniform diffusion. Seeing the intersection without the extrapolation is crucial to make such a claim. The paper itself states that as the model converges, the shape of the ELBO might change.

*W2* Another weakness is that Figure 2 does not seem to support the claims in the paper. The optimal batch size seems very much dependent on the model size, as x in $D^x$ seems to be different for different batch sizes. In particular it seems to be smaller for larger models. Can the authors provide the values of x for each model size and diffusion type? I believe that the x will vary a lot with respect to 0.8179.

*W3* A similar comment holds true for the optimal learning rate with respect to batch size. Can you please provide the exponents for each model size/diffusion type?

**Other weaknesses**

*W4* Some experiments seem to be unfinished at the time of submission. Given that ICLR allows updating the manuscript during the rebuttal period however, this should be rectifiable.

*W5* While the paper positions itself properly and cites related work, some comparisons are missing. How does the formulation in terms of SNR compare with that of [1]? That paper also presents a similar result in the case of masked diffusion.
Regarding the training without the weighting term, something very similar was originally presented in [2] in the form of CEDD which removes the time-weighting term from the loss, and CEDD* which finds and even more appropriate weighting compared to the usual CEDDT. The same paper also presents a similar diffusion process to the hybrid in this paper and in [3], which they call roulette and which predates GIDD. How does that compare to the hybrid model in this paper?

[1] Sahoo et al. Simple and Effective Masked Diffusion Language Models. NeurIPS 2024

[2] Haxholli et al. Efficient Perplexity Bound and Ratio Matching in Discrete Diffusion Language Models. ICLR 2025.

[3] Rutte et al. Generalized Interpolating Discrete Diffusion. ICML 2025.

**Questions:**

**Q1** Would it be possible to extend the curves in Figure 4 for the uniform and masked one? Ideally up to 15x Flops? (if not possible at least 2-3x flops so that we can check if the extrapolation is appropriate).

**Q2** Can the authors provide the values of exponents for each model size and diffusion type regarding Figure 3 (both sides)?

**Q3** Would it be possible to add the ALM results in Figure 4 for completeness?

**Q4** The larger optimal batch sizes hold under the sub-epoch training assumption. Is it reasonable to make this assumption in practice?

---

> ### Author Response · Authors · 2025-11-26
> **Official response to Reviewer foon (1/2)**
>
> We thank the reviewer for their insightful feedback and for acknowledging the importance of advancing the knowledge on scaling diffusion models. In the following we will address the reviewer’s concerns.
>
> **W1.** The reviewer raises concerns about the claim that uniform diffusion scales more favorably in compute-bound settings. We agree with the reviewer that this is a strong claim that ideally should be verified empirically. To this end, we have trained scaled-up versions of our masked and uniform diffusion models up to 10^21 training FLOPs (3B parameters) in addition to a 10B parameter uniform diffusion model trained on 10^22 FLOPs (5x and 50x larger than our previously largest runs). Due to computational constraints we were unfortunately unable to train an additional 10B parameter masked diffusion model. Simultaneously, we have refined our methodology for estimating the scaling laws, switching from Approach 3 by Hoffmann et al. [4] (parametric loss surface fit) to Approach 2 (iso-FLOP curves), which is more widely used in the literature [5,6] and produces a better fit to our observations. Our scaled-up 3B and 10B parameter runs closely match the predictions from this better fit (obtained from runs up to 2e20 FLOPs) and confirm that the likelihood gap between masking and uniform noise shrinks with increased training compute (from 3.2% at 10^18 FLOPs to only 1.7% at 10^21 FLOPs).
>
> While the scaled-up runs closely match the predictions, the fitted scaling laws have shifted towards more convergent behavior between different noise types in the compute-bound regime. This does shift the final conclusion: It now appears that all noise types can scale equally well given enough training compute while being competitive with, or potentially even scaling better than, ALMs in compute-bound settings (https://ibb.co/gZ1k28LP; ALM scaling laws by [5]). Please see our general comment for further details.
>
> **W2, W3, Q2.** The reviewer raises concerns about the optimal batch size scaling, specifically how the scaling exponent may be different for different model sizes. We agree that this is an important aspect to check, and provide scaling exponents grouped by both noise type and model size for both optimal batch size and optimal learning rate (as a function of optimal batch size).
>
> Optimal batch size $B^*$:
> | Group | Slope (±99% CI) | R² |
> |-|-|-|
> | all | 0.8225 ± 0.0104 | 0.9754 |
> ||||
> | masked| 0.7759 ± 0.0167 | 0.9846 |
> | low-uniform | 0.7916 ± 0.0194 | 0.9811 |
> | balanced | 0.8373 ± 0.0214 | 0.9806 |
> | high-uniform | 0.8607 ± 0.0206 | 0.9832 |
> | uniform | 0.8787 ± 0.0160 | 0.9899 |
> ||||
> | 25.2M | 0.7847 ± 0.0202 | 0.9838 |
> | 49.2M | 0.8148 ± 0.0220 | 0.9806 |
> | 85.1M | 0.8395 ± 0.0180 | 0.9854 |
> | 201.6M | 0.8127 ± 0.0175 | 0.9840 |
> | 566.7M | 0.8365 ± 0.0163 | 0.9861 |
>
> Optimal learning rate $\eta^*$:
> | Group | Slope (±99% CI) | R² |
> |-|-|-|
> | all | 0.3412 ± 0.0110 | 0.9089 |
> ||||
> | masked | 0.3295 ± 0.0286 | 0.8846 |
> | low-uniform | 0.2924 ± 0.0290 | 0.8360 |
> | balanced | 0.3653 ± 0.0207 | 0.9437 |
> | high-uniform | 0.3628 ± 0.0190 | 0.9489 |
> | uniform | 0.3554 ± 0.0151 | 0.9643 |
> ||||
> | 25.2M | 0.3422 ± 0.0202 | 0.9326 |
> | 49.2M | 0.4606 ± 0.0637 | 0.8286 |
> | 85.1M | 0.3293 ± 0.0193 | 0.9235 |
> | 201.6M | 0.3279 ± 0.0158 | 0.9449 |
> | 566.7M | 0.3340 ± 0.0327 | 0.8729 |
>
> We find that while there are slight fluctuations in the scaling exponents grouped by model size, there is no consistent pattern, with 99% confidence intervals often overlapping. More data would be needed to form a definitive answer, but based on the available observations we argue that the optimal batch size and learning rate are, more likely than not, unaffected by model size. Note that this independence of learning rate to model size is an expected consequence of the CompleteP parameterization, which enables learning rate transfer across width and depth [8].
> On the other hand, grouping by noise type reveals a consistent pattern of more uniform noise favoring larger batches. That said, the difference is rather small and unlikely to have practical implications as batch sizes are commonly rounded to a power of two for computational efficiency. We will add these results to the manuscript and ensure that the framing accurately represents our results (e.g. L120 currently claims that the optimal batch size is “independent of model size and noise type” which is too strong of a statement given the data).
>
> **W4.** The writing has been updated with numbers based on the complete hyperparameter sweep and using the improved methodology (Hoffmann et al. [4], Approach 2) and will be uploaded as soon as possible. Note that the numbers in the previous paragraph may deviate slightly from the ones in the original version since they include the complete sweep.
>
> [continued]

---

> ### Author Response · Authors · 2025-11-26
> **Official response to Reviewer foon (2/2)**
>
> **W5.** Regarding the prior SNR formulations of the masked diffusion ELBO proposed by [1] and [7], we would like to highlight that our proposition is more general than that. It extends beyond masked diffusion and encompasses all interpolating diffusion processes with a smooth mixing schedule, in particular including masking, uniform and hybrid-noise diffusion. That said, we agree with the reviewer that the relevant work [1,7] needs to be cited and will do so in the revised version. Regarding [2], we were unaware of this work and agree that it is indeed relevant. We will make sure to properly cite it in the revised version. The main difference to [2] is that CEDD-roulette uses a time-constant mixing distribution, whereas we use a time-varying distribution that smoothly interpolates from uniform to masking noise, similar to GIDD [3].
>
> **Q1, Q3.** We have updated the plot in question with the updated scaling laws and scaled-up runs while also adding the scaling law for ALMs proposed by [5] for reference (https://ibb.co/gZ1k28LP). Note that absolute numbers are not directly comparable between our models and those of [5] since they are not trained on the same data.
>
> **Q4.** Indeed, the proposed values of the optimal batch size are likely not to generalize to multi-epoch training regimes where overfitting becomes a concern. However, we are considering internet-scale pre-training, where training corpora can consist of trillions of tokens. Generally speaking, doing more than a single pass on the data is prohibitive and not common practice. We will make sure to clarify this in the revised manuscript so as to avoid any potential misinterpretation.
>
> We hope that we were able to adequately address the reviewer’s concerns and invite them to point out any remaining issues so that they can be fixed in future revisions of the paper. If we were indeed able to address the reviewer’s core concerns, we kindly ask them to consider increasing their score.
>
> ---
>
> - [1] Sahoo et al. Simple and Effective Masked Diffusion Language Models. NeurIPS 2024
> - [2] Haxholli et al. Efficient Perplexity Bound and Ratio Matching in Discrete Diffusion Language Models. ICLR 2025.
> - [3] Rutte et al. Generalized Interpolating Discrete Diffusion. ICML 2025.
> - [4] Hoffmann, J., Borgeaud, S., Mensch, A., Buchatskaya, E., Cai, T., Rutherford, E., ... & Sifre, L. (2022). _Training compute-optimal large language models._ arXiv preprint arXiv:2203.15556.
> - [5] Bi, X., Chen, D., Chen, G., Chen, S., Dai, D., Deng, C., ... & Zou, Y. (2024). _Deepseek LLM: Scaling open-source language models with longtermism._ arXiv preprint arXiv:2401.02954.
> - [6] Grattafiori, A., Dubey, A., Jauhri, A., Pandey, A., Kadian, A., Al-Dahle, A., ... & Vasic, P. (2024). _The Llama 3 herd of models._ arXiv preprint arXiv:2407.21783.
> - [7] Shi, J., Han, K., Wang, Z., Doucet, A., & Titsias, M. (2024). _Simplified and generalized masked diffusion for discrete data._ Advances in neural information processing systems, 37, 103131-103167.
> - [8] Dey, N., Zhang, B. C., Noci, L., Li, M., Bordelon, B., Bergsma, S., ... & Hestness, J. (2025). _Don't be lazy: CompleteP enables compute-efficient deep transformers._ arXiv preprint arXiv:2505.01618.

---

### Official Review · Reviewer_5DFM · 2025-10-31

**Soundness:** 2
**Presentation:** 1
**Contribution:** 2
**Rating:** 2
**Confidence:** 3

**Summary:**

This paper focuses on the scaling characteristics of DLMs, examining their performance across varied noise types. To do this, the author proposed a framework to unify the parameterization of uniform and masked diffusion models, which allows for smooth transitions between the two models. This paper investigates the scaling behavior of DLMs with respect to critical hyperparameters, including batch size, learning rate.

**Strengths:**

It is an insightful idea to unify the formulations of uniform and masked diffusion models via reparameterizing Signal-Noise-Ratio (SNR), which could help to explore new types of diffusion models in future work.

**Weaknesses:**

1. The authors sometimes made very strong claims without proper scientific support. For example, in Line#108, “... showing that training with and without annealing yields similar optima and a similar loss, up to some constant factor”. However, it turns out that the authors only compared the shapes of the loss curves of two different models through visual inspection, without any statistical analysis (Line#361 and 362).

2.The analysis of optimal batch size and optimal step count is confusing. In Line#370 and Eq(14), the authors introduced 4 variables, i.e., step count S, batch size B, optimal learning rate and the same observed loss L, but in the following equation (Eq 14), only S and B appear in the target fitting function. Please can you elaborate?

3. The conclusions about optimal batch size presented appear to be questionable. Specifically, in Line 397, the authors state that the optimal batch size (B) which can grow up to 2^10 x 10^9.9. This finding is presented as "in stark contrast" to existing literature, which suggests that smaller batch sizes generally lead to improved test accuracy. However, the authors also acknowledge that this discrepancy might stem from the fact that other research considers overfitting, whereas their work assumes overfitting is not a factor which seems unrealistic in practice. This raises a question regarding the reliability of their conclusion.

The presentation of the paper could be improved. I’m listing a few examples:

1.Figure 1, which reports the key findings in the paper, is hard to navigate, and more explanations in the caption could improve the presentation.

2. In Line#073&074, it’s seems overstated to say “strictly more difficult”, as the difficulty can depend on specific tasks. In addition, some theoretical analysis to support this will be desirable.

3. In the legend of Fig2 (a) (b), I assume “cd” stands for cooldown? If so, you should define the abbreviation in the caption.

4. In Fig 2(c), neither of the model labels L8-D512 and L12-D768 is formally introduced in the main text, and they only appear silently in Table 2 in the Appendix.

**Questions:**

1. In Fig2, why do the ELBOs decrease over the training course? Are you meant to say negative log-likelihood or loss?

2. In Fig2(c), you compared the shape of the annealed loss with the unannealed loss. Please can you clarify which curve corresponds to what?

3. In Line#364, you claimed that you “do not expect the conclusions to change depending on the noise type.” Please could you elaborate more on this conclusion?

4. In Line#377, what do you mean by “token-optimal batch size”? You mean token-size-optimal batch size?

5. As the definitions of P (non-embedding parameters), D (total #training tokens), and loss (L) are scattered in different sections, while Table 1 summarizes all these metrics, it’ll be much more accessible to the reader if Table 1 can explain these symbols in the caption.

---

> ### Author Response · Authors · 2025-11-26
> **Official response to Reviewer 5DFM (1/2)**
>
> We thank the reviewer for their insightful and constructive feedback. In the following, we address the points that were raised.
>
> ### W1. Learning rate annealing
> The reviewer raises concerns about the statistical significance of our claim that learning rate annealing brings a constant improvement compared to training without learning rate cooldown.
> We agree with the reviewer that the presentation of our annealing ablations was not sufficiently detailed and we would like to provide additional numerical results.
> Adopting a constant-factor improvement hypothesis, the ratio of unannealed loss to annealed loss is $1.0245 \pm 0.00138$ (99% confidence interval, obtained through normal approximation). Furthermore, a linear fit ($R^2 = 0.988$) extrapolates remarkably well to the new scaled-up runs at 10^21 and 10^22 training FLOPs (https://ibb.co/dzQPgyd; the unannealed loss values of the scaled-up runs are obtained through log-log extrapolation from the most recent loss values before annealing begins). Altogether, we argue that this constitutes sufficient evidence for the claim that learning rate annealing provides a constant-factor improvement and will add the statistical analysis provided here to the updated version of the paper.
>
> ### W2, W3. Relation between batch size and step count (Section 4.2)
> **W2.** We agree with the reviewer that the explanation of Eq. 14, as well as the quantities introduced in its context, were confusing, and we would like to clarify.
> The equation is a way to characterize the data we observe through the following procedure.
> 1. For any target loss L
> 2. Iterate through all batch sizes B, assume that the LR is well-tuned
> 3. For all batch sizes B, observe the step count S at which the target loss L is reached.
> 4. Plotting the observed points (B, S), we observe a hyperbolic relationship which is stated in Eq. 14.
>
> Therefore the final hyperbola only depends explicitly on B and S and implicitly (through the constants) on L.
> Please refer to Fig. 5 (App. A.1) for an example result of this procedure, along with the hyperbolic fit.
> This result is purely phenomenological: It accurately describes our observations, but does necessarily provide further insights into the mechanism(s) underlying them. We have updated the corresponding paragraph for improved clarity to hopefully prevent any future confusion.
>
> **W3.** We agree with the reviewer that the way the paragraph was written lacked clarity. While we don’t necessarily know the underlying cause for this hyperbolic relation, it suggests that there exists an effective minimum batch size $B_{\min}(L)$ required to reach a certain target loss $L$ as well as an optimal batch size $B^\*(L)$ and step size $S^\*(L)$ that minimize some token budget $D = BS$. It is important to note that these statements are entirely about pre-training loss and compute efficiency, not about test accuracy.
> In the original draft, we used a power law to fit the observed values of $B_{\min}(L)$ and $S_{\min}(L)$ and explicitly stated their extrapolated values for $L \to 0$. While we did mention in a footnote that the power law approximation of $B_{\min}(L)$ and $S_{\min}(L)$ likely breaks down as the loss approaches it minimum achievable value, we do agree that these extrapolated values understandably gave the impression that we are claiming enormous values for the optimal batch size, which was not our intention. Consequently, we have removed these numerical extrapolations in the revised manuscript.
> On first glance, the mere existence of a minimum batch size required to achieve some target loss may still appear to contradict classical work showing that small batches benefit generalization (colloquially, people often claim that “the optimal batch size is 1”). This tension can be resolved by recognizing that we are operating in a different training regime: Internet-scale pre-training operates in the sub-epoch regime, whereas the cited theoretical results assume arbitrarily many passes over the data.
>
> ### W4. Presentation of the paper
> We thank the reviewer for valuable feedback on how to improve the presentation of the paper. We agree that the suggested changes will help the paper and will make sure to implement them in the revised version. Regarding uniform diffusion being “strictly more difficult” than masked diffusion, this is based on the following informal argument: Consider a hypothetical uniform diffusion scenario where we are given additional information about which tokens are noisy and which are noise free. In this case, the denoising problem becomes equivalent to masked diffusion as it suffices to fill in the noisy (missing) tokens.
> Recent work by Amin et al. [1] provides a more detailed and formal analysis of this intuition, so we will make sure to reference it in the revised manuscript to support our informal argument.
>
> [continued]

---

> ### Author Response · Authors · 2025-11-26
> **Official response to Reviewer 5DFM (2/2)**
>
> Questions:
> 1. Indeed, we slightly abuse notation throughout the paper and often use the term “ELBO” to refer to the negative ELBO, which is an upper bound on the NLL (note that the ELBO is always negative whereas the negative ELBO is always positive). This is done for consistency and easy comparison with the ALM literature, where the loss is usually stated in terms of NLL. We will make sure to clarify this convention so as to avoid confusion.
> 2. This figure overlays the training loss of runs both with and without annealing, where the annealed runs “branch” off the unannealed run at 4 distinct points. The final loss values of the various annealed runs are plotted as dots/squares whereas the unannealed loss is drawn in the dashed line. We agree that the clarity of this plot could be improved and will rework it for a future version of the paper.
> 3. The conclusion here would be that learning rate annealing brings a constant loss improvement (of around 2.5%). While we did not explicitly verify this on all noise types (masked, uniform, hybrid-noise), we argue that it is reasonable to assume that it will generalize to other noise types given the strength of the correlation.
> 4. Token-optimal batch size refers to optimality in the number of training tokens: For a given budget of training tokens, which batch size achieves the lowest training loss? This batch size is considered (token-)optimal. Note that token-optimality implies compute-optimality if the model size is fixed.
> 5. This is a good idea, and we agree that it would improve the clarity of Table 1. We will add it to the revised version.
>
> We hope that we were able to adequately address the reviewer's concerns and invite them to share any remaining or additional points, and to provide further feedback on how the manuscript could be improved.
>
> ---
>
> [1] Amin, A. N., Gruver, N., & Wilson, A. G. (2025). _Why Masking Diffusion Works: Condition on the Jump Schedule for Improved Discrete Diffusion._ arXiv preprint arXiv:2506.08316.

---

### Official Review · Reviewer_a8ML · 2025-11-01

**Soundness:** 2
**Presentation:** 3
**Contribution:** 3
**Rating:** 4
**Confidence:** 3

**Summary:**

This work conducts various analyses related to the scaling laws of diffusion language models. They cast the diffusion objective into a form where the noise schedule is based on a SNR ratio and then conduct multiple analysis on how the training ELBO scales w.r.t parmater count, token budgets, and various hyper parameters.

**Strengths:**

This work is well presented and covers a reasonable range of noise schedules, model sizes, and various hyper parameters. Scaling models are quite important to guide future work and this is work is executed well enough to be generally impactful and useful. While I have some concerns regarding some evals I believe are missing, I believe this work could be a timely and useful addition to the community if these shortcomings are addressed.

**Weaknesses:**

This work is mainly limited by only presented results in terms of the ELBO and not performing any sort of downstream evaluation. This limits any comparisons to ALMs and allows for possible confounding where different mixing distributions during training may have different performance characteristics during downstream evaluation. The GIDD work this work cites a fair bit provides a reasonable set-up to perform downstream eval that would benefit this work immensely. In particular, one thing I would really like to see is how these scaling laws interact with different de-noising processes (e.g. LLaDa style decoders vs re-masking style decoders like ReMDM).

Additions such as these would significantly strengthen this work and in my opinion would clearly push this work into the accept range.

**Questions:**

What does some of the downstream performance looks like for some of the models you trained? Does Train ELBO correlate well with downstream performance? Do these comparisons hold across different types of de-noising processes?

---

> ### Author Response · Authors · 2025-11-26
> **Official response to Reviewer a8ML**
>
> We thank the reviewer for their insightful feedback, for pointing out the importance of the topic, and for acknowledging that our work is well-executed and generally impactful. In the following, we will attempt to address the reviewer’s concerns around missing evals.
>
> As requested by Reviewers ZRsL and foon, we have trained scaled-up versions of our models, obtaining 3B param. masked and uniform diffusion models trained for 10^21 FLOPs in addition to a 10B param. uniform diffusion model trained for 10^22 FLOPs. As these models are most comparable to other scaling literature, we focus the downstream performance evaluation on these scaled-up checkpoints. See our general comment for benchmark accuracies and further details. To summarize, and to answer the reviewer’s questions: While downstream performance generally correlates with training loss, there appear to be some discrepancies between knowledge-heavy tasks and reasoning-heavy tasks, with masked diffusion favoring the former and uniform diffusion favoring the latter.
>
> The reviewer also points towards comparing different sampling algorithms (“LLaDa style decoders vs re-masking style decoders like ReMDM”), which we agree is an interesting idea. Since most non-trivial sampling methods have only been proposed in the context of masked diffusion and do not trivially generalize to uniform diffusion, we developed a confidence-based decoding technique that generalizes to both masked and uniform diffusion and can be seen as an extension of “LLaDA-style decoding” to uniform noise. Specifically, we replace the standard confidence heuristic $\mathrm{conf}(z_t) = \delta_{z_t, m} \max_{z’} p_\theta(z’ | z_t)$ with a generalized version $\mathrm{conf}(z_t) = (\max_{z’} p_\theta(z’ | z_t) - p_\theta(z_t | z_t)) \cdot p_1(z_t)$, which reduces to the standard heuristic in the case of masked diffusion while also being applicable to uniform diffusion. Compared to standard ancestral sampling, this adaptive sampling technique gives a noticeable relative boost on GSM8k (max. generation length is set to 128, $T$ denotes the number of denoising steps):
>
> |Model|Ancestral ($T=256$)|Adaptive ($T=128$)|Adaptive ($T=256$)|
> |-|-|-|-|
> |mask-3b|1.06|1.67|n/a|
> |unif-3b|1.44|1.97|2.05|
> |unif-10b|2.12|2.27|2.43|
>
> While the accuracy is comparatively low, we note that this is a few-shot (k=4) setup without fine-tuning and that our pre-training contains no dedicated math or coding data. We also see that uniform diffusion can benefit from using more denoising steps than generated tokens even in an adaptive inference setting, which is impossible for masked diffusion without remasking.
>
> We hope that these downstream evaluations can adequately address the reviewer’s concern and invite them to point out if there are still any missing evaluations that may hinder the paper’s acceptance. We will be happy to add these as soon as possible, at the latest for the camera-ready version.

---

### Official Review · Reviewer_ZRsL · 2025-11-03

**Soundness:** 2
**Presentation:** 2
**Contribution:** 3
**Rating:** 4
**Confidence:** 4

**Summary:**

The paper presents a comprehensive study on the scaling behavior of discrete diffusion language models. Notably, the fitted scaling law comparing masked diffusion and uniform diffusion predicts an advantage of uniform diffusion in the >10^22 flops regime.

**Strengths:**

* This work studies an important topic that is of interest to the community working on discrete diffusion models for language. The methodology taken is generally sound except a few design choices explained below.

* Some experiment design choices are justified through ablations, e.g., the omission of annealing phase to save a search dimension.

* The conclusion drawn from the fitted scaling law is surprising and if verified, could lead to a paradigm shift in the discrete diffusion community. It is widely believed that uniform diffusion underperforms masked diffusion but this work suggest that uniform makes better use of the flops and might catch up and lead as we further scale.

**Weaknesses:**

* The writing needs significant improvement - currently the paper is lacking proper introduction of background materials - just citing them is not sufficient. For example, “ To support both isotropic and anisotropic denoising, we implement diffusion forcing (Chen et al., 2024) by sampling independent per-token noise levels for 50% of samples” the authors need to define what is “isotropic” and “anisotropic” denoising and introducing the diffusion forcing method. It is not immediately clear why such algorithmic choices are being made and how that influences the scaling laws. The derivation of the objective is also assuming familiarity with GIDD method, e.g., it is never defined what “q(x)_z” means in equation (3).

* The use of unweighted surrogate loss also seems concerning - in fact, it creates a train-test mismatch (ELBO is measured at test time). Does this choice significantly change the scaling behavior of all models? What happens if you use the commonly employed ELBO objective for training?

* The experiments are conducted on models up to 570M parameters and used at most \~10^20 flops. The paper's primary claims (uniform scales better than masking), however, are based on an extrapolation into 10^21\~10^22 flops based on a slope difference that is very small. The authors themselves admit the fit can be "brittle" and "sensitive to slight changes in the scaling exponents". Therefore, it is unclear whether the prediction would hold in practice. Recently there has been theoretical arguments made about the advantage of masked diffusion over uniform diffusion [4], which also seems to contradicting the prediction made here.

* Discrete diffusion is a fast growing field and many very related work is missing. The authors should at least cite the recent simplification of masked diffusion models [1, 2, 3]. Some of these works might reduce the contribution claimed in this work, e.g., [2] proposed the same SNR formulation alpha/(1 - alpha) which is claimed as a novel contribution in this work. Also, the connection between masked diffusion and uniform diffusion is extensively studied in [4], with conclusion contradicting the one drawn from scaling law fitted in this work. The authors are expected to have at least a discussion on this.

[1] Ou, J., Nie, S., Xue, K., Zhu, F., Sun, J., Li, Z., & Li, C. (2024). Your absorbing discrete diffusion secretly models the conditional distributions of clean data. arXiv preprint arXiv:2406.03736.

[2] Shi, J., Han, K., Wang, Z., Doucet, A., & Titsias, M. (2024). Simplified and generalized masked diffusion for discrete data. Advances in neural information processing systems, 37, 103131-103167.

[3] Sahoo, S., Arriola, M., Schiff, Y., Gokaslan, A., Marroquin, E., Chiu, J., ... & Kuleshov, V. (2024). Simple and effective masked diffusion language models. Advances in Neural Information Processing Systems, 37, 130136-130184.

[4] Amin, A. N., Gruver, N., & Wilson, A. G. (2025). Why Masking Diffusion Works: Condition on the Jump Schedule for Improved Discrete Diffusion. arXiv preprint arXiv:2506.08316.

**Questions:**

Please see above weaknesses.

---

> ### Author Response · Authors · 2025-11-26
> **Official response to Reviewer ZRSL (1/2)**
>
> We thank the reviewer for their extensive review and insightful feedback, as well as for acknowledging the importance of the topic and general soundness of our methodology, and for pointing out the potential for our results.
>
> ### Writing and presentation (W1, W4)
> The reviewer raises concerns about the writing: Lacking proper introduction of background material (diffusion forcing [5], GIDD [6]) and missing references to related prior work, in particular on masked diffusion [1,2,3] as well as theoretical results on inherent advantages of masked diffusion over uniform diffusion [4].
> We agree with the reviewer and thank them for the valuable feedback. While some of these omissions (e.g. introducing background material in more detail) were made due to space constraints, others are oversights on our end:
> - First, we agree that there is a noticeable lack of background material and that the paper assumes a lot of prior knowledge from the reader. We will add a proper introduction of background material, in particular on diffusion forcing [5] and GIDD [6].
> - While some missing prior work [1,2,3] is referenced indirectly via prior work on scaling masked diffusion [8], we agree that it should also be cited directly. We will make sure to add proper references in the revised version.
> - It is also true that SNR formulations of the masked diffusion ELBO have been proposed in prior work [2,3]. However, our proposition is more general than that. It extends beyond masked diffusion and encompasses all interpolating diffusion processes with a smooth mixing schedule, in particular including masking, uniform and hybrid-noise diffusion. That said, we agree with the reviewer that the relevant work [2,3] needs to be cited, and, upon reexamining our writing, that the framing of our contribution is not entirely transparent about the prior existence of these formulations. We will make sure to rectify this in the updated version.
> - Our findings do not contradict [4], and in fact, we argue that both papers provide compatible findings and explanations: Since uniform diffusion is a strictly more difficult task, it requires more capacity, resulting in a likelihood gap at small scales. Borrowing terminology from [4], small models struggle to learn both the transition schedule and the transitions themselves, but as we increase the size of the model this becomes less of an issue since there is enough capacity to learn both. This can also help explain why uniform diffusion calls for scaling model size more quickly than data (compared to masked diffusion). We will make sure to include this discussion in the revised version.
>
> ### Diffusion forcing (W1)
> The reviewer raises concerns about the choice of incorporating independent per-token noise, both that it is not introduced properly and that it may affect the scaling behavior. First, the terms “isotropic” and “anisotropic” are used somewhat informally, referring to whether the noise level is the same/even (-> isotropic) or different/uneven (-> anisotropic) across tokens. We will make sure to clearly define this terminology in the revised version.
>
> Regarding the motivation and implications of including per-token independent noise levels, it is first important to note that sampling global or per-token noise levels does not affect the per-token ELBO under the standard factorizing assumption ($p_\theta(z_s^{(1:L)} | z_t^{(1:L)}) = \prod_{i=1}^L p_\theta(z_s^{(i)} | z_t^{(1:L)})$) and simply corresponds to switching the order of taking the expectation over $t$ and the sum over $L$. All quantities are therefore valid likelihood bounds of the data and there are _no further implications_ on the reported numbers and scaling behavior. Anecdotally, we find that anisotropic noise slightly reduces variance and improves convergence, which is also corroborated by [5] (App. B.2). We agree that this is not immediately obvious and will make sure to elaborate on it in more detail in the revised version. The motivation for including diffusion forcing in our training largely follows [5] (App. B.2): stabilization of autoregressive rollout, modeling causal uncertainty, and generally enhanced flexibility in the denoising order. Concurrent work [7] also corroborates the advantages of diffusion forcing for discrete diffusion models, enabling increased throughput at minimal quality degradation. We will make sure to mention all of this in the revised version.
>
> ### Surrogate loss (W2)
> The reviewer raises concerns that the use of a surrogate loss (unweighted diffusion ELBO) may lead to a train-test mismatch. To clarify: all the reported numbers are in terms of the (negative) diffusion ELBO, which constitutes a true (upper) bound on the NLL and is directly comparable to other likelihood measures such as the CE loss of AR models, since continuous-time diffusion ELBOs are considered relatively tight [9].
>
> [continued in next comment]

---

> ### Author Response · Authors · 2025-11-26
> **Official response to Reviewer ZRsL (2/2)**
>
> ### Predictive strength of scaling laws (W3)
> We agree with the reviewer that the reliability and predictive strength of the reported scaling laws is a central question. A detailed response is provided in our general comment above. To summarize: We updated our scaling law estimation methodology to a more standard approach (based on ISO-FLOP curves instead of fitting the parametric loss surface), which produces a more stable fit with tighter confidence intervals. Our core claims still hold:
> We still find that uniform diffusion catches up to masked diffusion as we scale to large compute budgets. However, the scaling behavior of different noise types now appears to be more convergent, with no clear winner even at very large scales.
> Following the new scaling laws, we have also trained scaled up versions of both masked and uniform diffusion with 3B parameters at 10^21 FLOPs as well as a 10B uniform diffusion model at 10^22 FLOPs (5x and 50x larger than the largest ablation run), which closely follow the predicted scaling trends (https://ibb.co/gZ1k28LP; ALM scaling laws by [10]). Importantly, the observed likelihood gap between masked and uniform diffusion shrinks from 3.2% at 10^18 FLOPs to only 1.7% at 10^21 FLOPs.
>
> We hope that we were able to adequately address the concerns of the reviewer. We invite the reviewer to share any remaining issues so they can be addressed in future revisions. If we were indeed able to address the reviewer’s core concerns, we kindly ask them to consider increasing their score.
>
> ---
>
> - [1] Ou, J., Nie, S., Xue, K., Zhu, F., Sun, J., Li, Z., & Li, C. (2024). _Your absorbing discrete diffusion secretly models the conditional distributions of clean data._ arXiv preprint arXiv:2406.03736.
> - [2] Shi, J., Han, K., Wang, Z., Doucet, A., & Titsias, M. (2024). _Simplified and generalized masked diffusion for discrete data._ Advances in neural information processing systems, 37, 103131-103167.
> - [3] Sahoo, S., Arriola, M., Schiff, Y., Gokaslan, A., Marroquin, E., Chiu, J., ... & Kuleshov, V. (2024). _Simple and effective masked diffusion language models._ Advances in Neural Information Processing Systems, 37, 130136-130184.
> - [4] Amin, A. N., Gruver, N., & Wilson, A. G. (2025). _Why Masking Diffusion Works: Condition on the Jump Schedule for Improved Discrete Diffusion._ arXiv preprint arXiv:2506.08316.
> - [5] Chen, B., Martí Monsó, D., Du, Y., Simchowitz, M., Tedrake, R., & Sitzmann, V. (2024). _Diffusion forcing: Next-token prediction meets full-sequence diffusion._ Advances in Neural Information Processing Systems, 37, 24081-24125.
> - [6] von Rütte, D., Fluri, J., Ding, Y., Orvieto, A., Schölkopf, B., & Hofmann, T. (2025). _Generalized interpolating discrete diffusion._ arXiv preprint arXiv:2503.04482.
> - [7] Wang, X., Xu, C., Jin, Y., Jin, J., Zhang, H., & Deng, Z. (2025). _Diffusion LLMs can do faster-than-AR inference via discrete diffusion forcing._ arXiv preprint arXiv:2508.09192.
> - [8] Nie, S., Zhu, F., Du, C., Pang, T., Liu, Q., Zeng, G., ... & Li, C. (2024). _Scaling up masked diffusion models on text._ arXiv preprint arXiv:2410.18514.
> - [9] Kingma, D., Salimans, T., Poole, B., & Ho, J. (2021). _Variational diffusion models._ Advances in neural information processing systems, 34, 21696-21707.
> - [10] Bi, X., Chen, D., Chen, G., Chen, S., Dai, D., Deng, C., ... & Zou, Y. (2024). _Deepseek LLM: Scaling open-source language models with longtermism._ arXiv preprint arXiv:2401.02954.

---

### Author Response · Authors · 2025-11-26
**General Comment**

We would like to express our appreciation to all reviewers for taking the time to give constructive and insightful feedback. In this comment, we would like to address some common concerns and also shed light on some updates when scaling up (as also requested by reviewers) after the paper submission. The updated manuscript will be uploaded in the coming days, incorporating all the changes outlined below.

First, we have refined our methodology for estimating the scaling laws. While we originally followed Approach 3 from Hoffmann et al. [1] (parametric fit of the loss surface), this was producing suboptimal fits to the data. We therefore switched to Approach 2 (iso-FLOP curves), which is more widely used in the literature [3,4] and produces a more reliable fit that explains the data better.
In particular, the new approach results in different scaling exponents and tighter confidence bounds (see the updated version of the paper for details). Nevertheless, our original findings remain largely unchanged:
1. The scaling behavior between noise types still differs significantly, with uniform diffusion still calling for less tokens-per-parameter than other noise types.
2. In the data-bound scaling regime, uniform diffusion still retains a notable advantage starting at a data budget of ~200B tokens.
3. However, in terms of compute-bound scaling, different noise types are converging to very similar loss values for growing compute budgets, with uniform diffusion retaining only a slight advantage.
4. On the other hand, our results now indicate that all examined diffusion types have the potential to match or even surpass the performance of ALMs at scale, still supporting the case for DLMs being a viable alternative to ALMs.

Overall, this somewhat weakens our claim that the picture is “exceedingly favorable for uniform diffusion”, since in the compute-bound scaling regime there now is no clear advantage for one noise type over another. We have updated the paper accordingly, adding the updated methodology and scaling exponents while also updating the framing in the introduction and conclusion to accurately characterize our results.

Reviewers have also expressed interest in scaled up versions of our models (ZRsL, foon) as well as downstream evaluations (a8ML) in order to validate the predicted scaling behavior and comparing our results with prior work on scaling laws of DLMs and ALMs. To this end, we have trained masked and uniform diffusion models with 3B parameters on 10^21 training flops in addition to a 10B parameter uniform diffusion model trained on 10^22 FLOPs (5x and 50x larger, respectively, than our previously largest run). The observed losses closely match the predicted values and demonstrate that DLMs are able to keep up with ALMs at scale (https://ibb.co/gZ1k28LP; note that DeepSeek [3] uses a different dataset, making absolute loss numbers difficult to compare directly). We also evaluate the scaled-up checkpoints on a range of standard NLP benchmarks:

|Model|ARC-E|ARC-C|WinoG|PIQA|OBQA|BoolQ|GSM8k|
|-|-|-|-|-|-|-|-|
|mask-3b|49.9|29.4|51.6|64.8|30.6|60.9|1.67|
|unif-3b|50.6|29.4|51.1|63.5|28.8|56.4|2.05|
|unif-10b|61.8|35.7|55.5|66.3|32.8|60.3|2.43|

While benchmark accuracies generally correlate with training ELBO, there seems to be an interesting pattern of masked diffusion performing comparatively better on knowledge-heavy tasks (PIQA, OBQA, BoolQ) and uniform diffusion performing better on reasoning-heavy tasks (ARC-E, ARC-C, GSM8k). The relatively low performance on GSM8k may be explained by a lack of math and coding data in our training set, which is corroborated by a pass rate of 0% on HumanEval by all three checkpoints.

**Evaluation details**

Except for GSM8k, these benchmarks are multiple-choice questions where the answer is selected based on likelihood. We report the best accuracy between 128/256 denoising steps and fill any unused context with tokens sampled from the prior distribution. For GSM8k, we employ a confidence-based sampling technique [2] with a generalized confidence heuristic that is applicable to both masked and uniform diffusion: $\mathrm{conf}(z_t) = (\max_{z’} p_\theta(z’ | z_t) - p_\theta(z_t | z_t)) \cdot p_1(z_t)$, i.e. the maximal model confidence minus the current confidence (select by highest potential improvement) multiplied by the prior probability of the current token (only update noisy tokens). See our reply to Reviewer a8ML for further details.

---

[1] Hoffmann et al. (2022). _Training compute-optimal large language models._ arXiv preprint arXiv:2203.15556.

[2] Kim et al. (2025) _Train for the worst, plan for the best: Understanding token ordering in masked diffusions._ arXiv preprint arXiv:2502.06768.

[3] Bi et al. (2024). _Deepseek LLM: Scaling open-source language models with longtermism._ arXiv preprint arXiv:2401.02954.

[4] Grattafiori et al. (2024). _The Llama 3 herd of models._ arXiv preprint arXiv:2407.21783.

---

### Author Response · Authors · 2025-11-28
**Updated Manuscript**

Dear reviewers, despite today's unfortunate news, we have updated the manuscript with all the requested changes.
For reference, we highlight some of the changes that we've made for the new version:
- Improved methodology and validation of the predicted scaling behavior up to $10^{22}$ FLOPs, along with additional details and numbers on the obtained scaling laws.
- Addition of scaled-up models with 3B and 10B parameters and their downstream performance on a range of standard NLP benchmarks.
- Additional detail on background and prior work. In particular, we have expanded the section on discrete diffusion and GIDD [1], added a new section on Diffusion Forcing [2], and added additional references to prior work on SNR formulations of the ELBO [3,4] and hybrid-noise diffusion [5].
- A more detailed statistical analysis of the learning rate annealing ablation study.
- Additional analysis of the scaling behavior of scaling parameters.
- Improved presentation throughout. In particular, we improved the clarity of Section 4.2 (which is now Section 4.5) as per Reviewer ZRSL's feedback.

We again thank the reviewers for their valuable time and constructive feedback, and we believe that the updated manuscript is much improved thanks to it.

---

[1] von Rütte, D., Fluri, J., Ding, Y., Orvieto, A., Schölkopf, B., & Hofmann, T. (2025). Generalized interpolating discrete diffusion. arXiv preprint arXiv:2503.04482.

[2] Chen, B., Martí Monsó, D., Du, Y., Simchowitz, M., Tedrake, R., & Sitzmann, V. (2024). Diffusion forcing: Next-token prediction meets full-sequence diffusion. Advances in Neural Information Processing Systems, 37, 24081-24125.

[3] Shi, J., Han, K., Wang, Z., Doucet, A., & Titsias, M. (2024). Simplified and generalized masked diffusion for discrete data. Advances in neural information processing systems, 37, 103131-103167.

[4] Sahoo, S., Arriola, M., Schiff, Y., Gokaslan, A., Marroquin, E., Chiu, J., ... & Kuleshov, V. (2024). Simple and effective masked diffusion language models. Advances in Neural Information Processing Systems, 37, 130136-130184.

[5] Haxholli et al. Efficient Perplexity Bound and Ratio Matching in Discrete Diffusion Language Models. ICLR 2025.

---

### Meta-Review · Area_Chair_GofH · 2026-01-14

**Summary:**

This paper studies the scaling behavior of discrete diffusion language models. Reviewers agreed that the topic is important. Initial reviews raised concerns about reliance on extrapolated scaling laws, presentation clarity, and missing evaluations. The authors’ rebuttal provided large-scale experiments and expanded evaluations.

**Reviewer Concerns:**

Reviewers expressed concerns about the reliability of the fitted scaling laws and the strength of the claims. Other concerns included insufficient background, confusing presentation of batch-size and learning-rate analyses, and lack of downstream evaluation. In the rebuttal, the authors added substantially larger training runs, included benchmark evaluations, and revised framing.

**Reviewer Scores:**

Scores initially were marginally below threshold, with one clear reject. The rebuttal addressed reviewers' main concern about additional experiments. It is likely that three out of four reviewers could have upgraded their scores, which would have pushed the paper towards acceptance.

---

### Decision · Program_Chairs · 2026-01-26

Accept (Poster)